# Model-Size Scaling Laws for Classification Networks via Manifold Geometry and Box-Model Dimension

## Abstract

Our primary goal is to explain scaling laws with respect to the number of parameters in deep neural networks (DNNs), under structural assumptions on the target classification boundary, the training data distribution, and manifold geometry. Our analysis suggests that the generalization error bound follows a linear trend in the log-log plane for networks of large parameter sizes. The rate of this scaling law can be captured by the *box-model dimension*, which reflects how the number of model parameters grows as the radius of covering balls decreases. We also present evidence that the box-model dimension tends to vary across learning methods and architectures.

As a secondary contribution, we present preliminary evidence that several standard network architectures with a modular design can exhibit a discrete scale-invariance property, which may help predict the generalization error of larger models within a wide range of model sizes. We illustrate this using published results for ResNet and VGG models trained on ImageNet, ViT models trained on ImageNet, and InceptionNet models trained on Udacity.

## 1 Introduction

Scaling laws in deep neural networks (DNNs) indicate that, across a wide range of architectures and tasks, the generalization error is primarily driven by scale. The generalization error typically decreases according to predictable power-law relationships associated with variables such as the number of parameters, the amount of training data, and the computational resources used for training.

While several theoretical results (Cortes et al. (1993); Bahri et al. (2024); Bisla et al. (2021)) show that the relationship between generalization error and sample size follows a scaling law, where the slope appears inversely proportional to the intrinsic dimension of the data manifold, a comparable theoretical account of scaling laws with respect to model size, and their connection to learning algorithms and network architecture, is still lacking. Understanding this relationship can motivate refinements to network and algorithm design; a simple prediction rule can also facilitate the deployment of larger models based on data derived from smaller ones.

Our analysis is motivated by the implication of the universal approximation theorem for shallow and deep neural networks (Cybenko, 1989): increasing the number of activation functions (and therefore the number of weights) improves the granularity of function approximation. This is further supported by results on the expressive power gained through depth specifically (Yarotsky, 2017). It is also motivated by the observation that a deep neural network with piecewise-linear activations partitions the input space into a finite collection of decision regions, within each of which the network's output behaves as an affine function. As additional layers are stacked, this partition is recursively refined: each layer subdivides the regions induced by the previous layers, allowing the network to represent increasingly complex decision boundaries with a number of regions that can grow exponentially in depth. The number of decision regions induced by a piecewise-linear classification network is closely related to the well-studied problem of counting linear regions of ReLU networks, for which Montufar et al. (2014) established foundational bounds as a function of depth and width, later tightened by Serra et al. (2018). To connect this analysis to the underlying data distribution and to make it tractable in high dimensions, we adopt the widely used manifold assumption in machine learning,

namely that the data distribution concentrates on a low-dimensional manifold embedded in the ambient input space; this assumption allows us to control the number of decision regions via a covering argument on the manifold, thereby circumventing the curse of dimensionality associated with the ambient dimension.

Our main goal is to provide a theoretical explanation of scaling laws with respect to model sizes. To this end, we study the manifold geometry of the decision regions induced by a one-hot classification network, under three assumptions to overcome the curse of dimensionality and control the generalization error. First, we assume dense coverage of the manifold $\mathcal{X}$ by the training points, allowing every point on the manifold to be well-approximated by a nearby training sample. Second, we assume bounded support of the data distribution: realistic data need not lie exactly on $\mathcal{X}$, so we allow the probability density $\mathcal{P}$ to occupy a thin neighborhood around it, with $\mathcal{P}$ bounded above and below on this neighborhood. Third, we assume either a histogram approximation or an interpolation condition: either sufficient training data are available within each decision region induced by the network $\mathcal{M}$ to accurately estimate the probability mass in that region, or $\mathcal{M}$ interpolates the training data exactly, i.e., the training error is zero.

Under these conditions, we show that the generalization error for a network is proportional to $\gamma_s$ when $\gamma_s$ is small, where $\gamma_s$ is the radius of the smallest covering balls of the decision regions induced by network $\mathcal{M}$. In contrast, for a network with large $\gamma_s$, this relationship is more complex (see Lemma 4). We then use the fact that a network with a larger number of parameters $\#W$ can generate a smaller $\gamma_s$ to relate the covering radius to the number of parameters, using an approach analogous to covering a shape with boxes of varying size. Here, rather than counting the number of boxes needed to cover the graph $(\mathbf{x}, f(\mathbf{x}))$, where $f$ is the target function, we count the number of parameters $\#W$ in the model derived by applying a learning algorithm to a network architecture, in order to obtain the covering of the graph across a range of covering radii. Applying this to Lemma 4, we obtain the relationship between the generalization error and the number of parameters $\#W$ of a network (Theorem 1). This leads to our definition of the box-model dimension, which characterizes the asymptotic behavior of this parameter-based covering in a manner resembling how fractal dimension characterizes the asymptotic behavior of the covering number.

As a secondary, exploratory contribution, without asymptotic analysis, we show that the box-model dimension is analytically tractable in a particular network structure, where large networks built from modular design blocks exhibit a discrete scale-invariance property. This manifests as a log-periodic pattern superimposed on a power-law trend in the log-log plot of generalization error against parameter count. This suggests a route toward robustly predicting generalization error for larger models. Such design regularities could inform the design of large-scale models by allowing scaling curves to forecast performance ahead of full-scale training. More accurate scaling-law predictions could influence decisions about training larger models, which carries compute and environmental costs. At the same time, by allowing researchers to estimate a model's performance before running very large training experiments, improved scaling predictions could also help reduce wasted compute from training runs that fall short of expectations. To illustrate this predictive potential, we provide preliminary numerical results based on published benchmark results for ResNet and VGG models trained on ImageNet, ViT models trained on ImageNet, and InceptionNet models trained on Udacity.

We discuss the limitations of our analysis, highlight the difficulty in numerically verifying it, and propose a future direction to enhance these theoretical results. We also admit that our illustrative examples are insufficient to fully justify the discrete scale-invariance property for prediction tasks. These points are discussed thoroughly in Section 6. Section 2 surveys some prior work. Section 3 establishes the connection between generalization error and manifold geometry via the covering radius. Section 4 proposes the box-model dimension and derives a scaling law with respect to model size. Section 5 shows that the generalization error of a large modular-architecture model can be predicted from smaller models, and provides illustrative examples using published data. Section 7 presents the conclusion.

## 2   Related works

Empirically, the generalization error of deep neural networks has been observed to follow a smooth power-law scaling of the form $aN^{-\alpha} + b$ with respect to dataset size $N$ (Hestness et al., 2017; Hutter, 2021), and analogous smooth power-law relationships have been identified with respect to model size and training steps

(Rosenfeld et al., 2019; 2021; Xiong et al., 2024). Under the assumption that these scaling laws are smooth and continuous, recent work has further shown that they can be used to reliably predict generalization error for larger models from observations on smaller ones (Choshen et al., 2024; Li et al., 2025). Extensive empirical evidence by Caballero et al. (2022) demonstrates that scaling laws can be piecewise continuous and can be predicted using a broken neural scaling law.

Some analysis aims to explain the scaling laws theoretically. Sharma and Kaplan (2022) propose a model explaining scaling laws in terms of training set size $D$ and parameter count $P$. Bahri et al. (2024) analyzes generalization error with respect to $D$ and $P$ across two regimes: one where both can grow unboundedly, and one where one is held fixed. Bisla et al. (2021) derives a closed-form expression for generalization error by approximating local density as uniform within hypercubes, obtaining results consistent with empirical scaling laws. Park et al. (2025) further shows that benign overfitting can occur in the classical regime for large $D$.

From studying tractable shallow networks, it is possible to derive the factors governing the underlying dynamic behaviour at different scales. Bartlett et al. (2020) shows that effective generalization requires the number of "unimportant" parameter directions to significantly exceed the sample size. Cui et al. (2021) bounds the generalization error of kernel ridge regression, showing that eigenvalue decay governs learning rates. Maloney et al. (2022) analyzes a solvable one-hidden-layer model in the limit of large data and many parameters, deriving analytical formulas linking spectral properties of the data to power-law scaling of test loss, and explaining why scaling laws can eventually plateau.

## 3 Geometry, sample density, and generalization error

### 3.1 Assumptions and simplifications

The generalization error measures the difference between the expected test error and the expected training error, while the generalization bound represents an upper bound of that error. The classification network $\mathcal{M}$ is responsible for learning the classifier function $f$. $\mathcal{M}$ is a piecewise constant function, dividing the input domain into non-overlapping decision regions and assigning each region a specific class. To understand the local geometry of the input of a neural network's function, we use covering balls in the input domain, focusing on two parameters: $\gamma_s$, the smallest radius such that every decision region can be covered by some ball of that radius, and $\beta_l$, the largest radius such that every decision region contains some ball of that radius. In Figure 1(a), $\gamma_s$ is the radius of the ball enclosing the larger of the two regions shown, while $\beta_l$ is the radius of the largest ball inscribed within the smaller of the two regions.

Our analysis makes three assumptions to overcome the curse of dimensionality of the original ambient input space and to control the generalization error bound.

In machine learning applications, typical of deep learning, the ambient dimension $D$ of an input is extremely large, making any covering argument over the ambient space intractable. To address this, we adopt the manifold assumption, which posits that the true data points lie on a smooth manifold of intrinsic dimension $d$ (Levina and Bickel, 2004; Ansuini et al., 2019). Let $\mathcal{X} \subset \mathbb{R}^D$ denote a compact, smooth $d$-dimensional submanifold of $\mathbb{R}^D$ (typically $d \ll D$, reflecting the low-dimensional structure of real-world data despite high ambient dimension). We write $\tau = \text{reach}(\mathcal{X}) > 0$ for its *reach*, i.e. the supremum of $r \geq 0$ such that every point within distance $r$ of $\mathcal{X}$ has a unique nearest point on $\mathcal{X}$. Suppose $\mathcal{X} \subseteq U_{\delta/2}(\mathcal{X})$ where $0 \leq \delta < \tau$. This allows us to sample over a narrow neighborhood of $\mathcal{X}$ to cope with noise in the distribution. When $\delta = 0$, the sample points are on $\mathcal{X}$.

**Assumption (dense coverage in the manifold)**: Assume $\delta/2 < \beta_l < \tau/2$. The training points form a $(\beta_l - \delta/2)$-covering of $\mathcal{X}$: for every $\mathbf{x} \in \mathcal{X}$, there exists a training point $\mathbf{x}_i$ with

$$\|\mathbf{x} - \mathbf{x}_i\|_2 \leq \beta_l - \frac{\delta}{2}. \tag{1}$$

Realistic data need not lie exactly on $\mathcal{X}$; we allow it to occupy a thin neighborhood around it.

**Assumption (bounded support)**: Fix a constant $0 \leq \delta < \tau$. The probability density $\mathcal{P}$ is supported on $U_{\delta/2}(\mathcal{X})$ and is bounded there: there exist constants $\infty > K \geq k > 0$ such that

$$\int_{\mathbb{R}^D \setminus U_{\delta/2}(\mathcal{X})} \mathcal{P}(\mathbf{x}) \, d\mathbf{x} = 0, \qquad k \leq \mathcal{P}(\mathbf{x}) \leq K \text{ for all } \mathbf{x} \in U_{\delta/2}(\mathcal{X}). \tag{2}$$

The last assumption controls how tightly the empirical risk tracks the true generalization error: without it, the decision regions induced by $\mathcal{M}$ could contain systematic mass imbalances invisible to a finite sample, inflating the gap between training and test performance in a way our bounds cannot account for.

**Assumption (histogram approximation or interpolation)**: We assume that one of the following two conditions holds: (i) sufficient training data are available for each decision region induced by the network $\mathcal{M}$ to allow an accurate approximation of the average probability mass within the region using a histogram over bins defined by the regions, or (ii) $\mathcal{M}$ is an interpolating network; i.e., the training error is zero. Note that condition (i) concerns the probability mass within each region $p$, i.e., $\int_p \mathcal{P}(\mathbf{x}) \, d\mathbf{x}$, not the pointwise density $\mathcal{P}(\mathbf{x})$.

We note that theoretically, it can be shown that under Assumption (bounded support), $\frac{n_p}{N}$ is an unbiased estimator of $\int_p \mathcal{P}(\mathbf{x}) \, d\mathbf{x}$ with variance tending to zero as $N \to \infty$ (Scott, 2015; Rudemo, 1982), where $n_p$ is the number of training points in region $p$ and $N$ is the number of training points in total.

Since $d \ll D$ in practice, expressing our results in terms of $d$ rather than $D$ yields substantially more favorable scaling and helps mitigate the curse of dimensionality. Formally justifying this substitution requires showing that covering arguments established over the ambient domain $\mathbb{R}^D$ remain valid when restricted to the data's intrinsic geometry; we establish this in the following two lemmas, which adapt standard tubular neighborhood and covering number results from Niyogi et al. (2008) to our setting with $\delta/2 < \beta_l < \tau/2$, and we discuss the geometric assumptions they rely upon.

**Lemma 1.** *Under the Assumption (bounded support) and Assumption (dense coverage in the manifold), the following hold:*

*(i) Genuine $\beta_l$-covering of the support tube: For every $\mathbf{y} \in U_{\delta/2}(\mathcal{X})$, there exists a training point $\mathbf{x}_i$ with*

$$\|\mathbf{y} - \mathbf{x}_i\|_2 \leq \beta_l. \tag{3}$$

*In particular, the training points form a $\beta_l$-covering of the support $U_{\delta/2}(\mathcal{X})$ of $\mathcal{P}$. When $\delta = 0$, this reduces to a $\beta_l$-covering of $\mathcal{X}$ itself, the sample points lying exactly on the manifold.*

*(ii) Well-defined projection: Since $\beta_l < \tau/2$ and $\delta < \tau$, the orthogonal projection $\pi_{\mathcal{X}} : U_{\tau/2}(\mathcal{X}) \to \mathcal{X}$ remains well-defined and unique throughout $U_{\delta/2}(\mathcal{X}) \subseteq U_{\tau/2}(\mathcal{X})$.*

*(iii) Intrinsic scaling of the covering number: The number of training points required satisfies*

$$N_{\mathrm{cov}}\left(\mathcal{X}, \beta_l - \delta/2, \|\cdot\|_2\right) \leq C \cdot \mathrm{vol}_d(\mathcal{X}) \cdot \left(\beta_l - \frac{\delta}{2}\right)^{-d} = \mathcal{O}(\beta_l^{-d}), \tag{4}$$

*where $C$ depends only on $d$, not on the ambient dimension $D$; the last equality holds for $\delta$ fixed as $\beta_l$ varies over $(\delta/2, \tau/2)$.*

*Proof.* Part (i). Let $\mathbf{y} \in U_{\delta/2}(\mathcal{X})$, so that $\|\mathbf{y} - \pi_{\mathcal{X}}(\mathbf{y})\|_2 \leq \delta/2$. By the Assumption (dense coverage in the manifold), there exists a training point $\mathbf{x}_i \in \mathcal{X}$ with $\|\pi_{\mathcal{X}}(\mathbf{y}) - \mathbf{x}_i\|_2 \leq \beta_l - \delta/2$. By the triangle inequality,

$$\|\mathbf{y} - \mathbf{x}_i\|_2 \leq \|\mathbf{y} - \pi_{\mathcal{X}}(\mathbf{y})\|_2 + \|\pi_{\mathcal{X}}(\mathbf{y}) - \mathbf{x}_i\|_2 \leq \frac{\delta}{2} + \left(\beta_l - \frac{\delta}{2}\right) = \beta_l. \tag{5}$$

By the Assumption (bounded support), $\mathcal{P}$ places no mass outside $U_{\delta/2}(\mathcal{X})$, so every point in the support of $\mathcal{P}$ is within $\beta_l$ of a training point. When $\delta = 0$, $U_{\delta/2}(\mathcal{X}) = \mathcal{X}$, and the bound reduces to the covering radius $\beta_l$ assumed directly on $\mathcal{X}$.

Part (ii). Immediate from $\beta_l < \tau/2$ together with the standard fact that the nearest-point projection onto a set of reach $\tau$ is well-defined and unique on $U_{\tau/2}(\mathcal{X})$ (Niyogi et al., 2008), and $U_{\delta/2}(\mathcal{X}) \subseteq U_{\tau/2}(\mathcal{X})$ since $\delta < \tau$.

Part (iii). By (Niyogi et al., 2008), since $\mathcal{X}$ is a compact $d$-dimensional submanifold with reach $\tau > 0$, its covering number at any radius $r < \tau/2$ satisfies $N_{\mathrm{cov}}(\mathcal{X}, r, \|\cdot\|_2) \leq C \cdot \mathrm{vol}_d(\mathcal{X}) \cdot r^{-d}$ for a constant $C$ depending only on $d$. Setting $r = \beta_l - \delta/2$, which lies in $(0, \tau/2)$ by the Assumption (dense coverage in the manifold), gives the stated bound. Since $\delta$ is fixed, $\beta_l - \delta/2 = \Theta(\beta_l)$ as $\beta_l$ ranges over $(\delta/2, \tau/2)$, so the bound is $\mathcal{O}(\beta_l^{-d})$. $\qquad\square$

Lemma 1 shows that the sampling complexity of the $\beta_l$-covering condition depends on the intrinsic dimension $d$ of the data manifold $\mathcal{X}$, not on the ambient dimension $D$.

The following lemma with proof given in Appendix A establishes (i) boundary alignment between the intrinsic and ambient classifiers, and (ii) that the generalization error integral reduces to a bounded-dimension covering argument. Assume $\gamma_s \leq \gamma_0/2$, the training data $\beta_l$-covers the unit $d$-ball (Lemma 3), the density $\mathcal{P}$ satisfies Assumption (bounded support) (with $\delta = 0$, per our simplification of notation), and the data satisfies Assumption (histogram approximation or interpolation). We let $\mathbf{1}_{\mathrm{hist}} \in \{0,1\}$ denote the condition indicator, where $\mathbf{1}_{\mathrm{hist}} = 1$ corresponds to condition (i) (histogram approximation) and $\mathbf{1}_{\mathrm{hist}} = 0$ to condition (ii) (interpolation).

**Lemma 2.** *Let $\mathcal{X} \subset \mathbb{R}^D$ be a compact $d$-dimensional submanifold with positive reach $\tau > 0$. Let $f^* : \mathcal{X} \to \mathbb{R}^l$ be an intrinsic classification function with decision boundary $\partial f^* \subset \mathcal{X}$, and let $f : U_{\tau/2}(\mathcal{X}) \to \mathbb{R}^l$ be its ambient extension defined by $f(\mathbf{y}) := f^*(\pi_{\mathcal{X}}(\mathbf{y}))$, where $\pi_{\mathcal{X}}$ is the nearest-point projection onto $\mathcal{X}$, which is well-defined and smooth for all $\mathbf{y} \in U_{\tau/2}(\mathcal{X})$. Suppose Assumption (bounded support) holds with parameter $\delta < \tau$. Then:*

*(i) Exact boundary alignment: The ambient decision boundary $\partial f$ restricted to $\mathcal{X}$ coincides exactly with the intrinsic boundary:*

$$\partial f \cap \mathcal{X} = \partial f^*. \tag{6}$$

*In particular, the quantities $\gamma_s$ and $\beta_l$ defined with respect to the decision regions of $f$ in $U_{\tau/2}(\mathcal{X})$ are identical to those defined with respect to the decision regions of $f^*$ in $\mathcal{X}$ (see proof below).*

*(ii) Dimension independence of risk: The generalization error of $f$ under $\mathcal{P}$ is entirely determined by its restriction to $U_{\delta/2}(\mathcal{X})$:*

$$\int_{\mathbb{R}^D} g(f(\mathbf{y}), \mathcal{M}(\mathbf{y})) \, \mathcal{P}(\mathbf{y}) \, d\mathbf{y} = \int_{U_{\delta/2}(\mathcal{X})} g(f(\mathbf{y}), \mathcal{M}(\mathbf{y})) \, \mathcal{P}(\mathbf{y}) \, d\mathbf{y}, \tag{7}$$

*where $g : \mathbb{R}^l \times \mathbb{R}^l \to \mathbb{R}_{\geq 0}$ denotes the loss function. Furthermore, all covering arguments required to bound the right-hand side scale as $\mathcal{O}(\beta_l^{-d})$ under the bounded-reach assumption, replacing the ambient scaling $\mathcal{O}(\beta_l^{-D})$.*

Together, Lemmas 1 and 2 justify the substitution $D \to d$ throughout all stated results, under the bounded-reach geometric assumption on $\mathcal{X}$ with $\delta/2 < \beta_l < \tau/2$. Specifically, Lemma 1 establishes that the training points act as an effective $\beta_l$-covering of the support tube $U_{\delta/2}(\mathcal{X})$, and that the number of training points required scales as $\mathcal{O}(\beta_l^{-d})$ rather than $\mathcal{O}(\beta_l^{-D})$. Part (i) of Lemma 2 ensures that the decision boundary geometry, and hence the quantities $\gamma_s$ and $\beta_l$, are preserved under the ambient extension $f$. Part (ii) of Lemma 2 establishes that the generalization error of $f$ under $\mathcal{P}$ is entirely determined by its restriction to $U_{\delta/2}(\mathcal{X})$, and that all covering arguments required to bound this error scale as $\mathcal{O}(\beta_l^{-d})$ under the bounded-reach assumption.

**Simplification of notation.** We assume $\mathcal{X}$ can be parametrized by the unit $d$-ball $B_d(\mathbf{0}, 1)$ via a smooth, bi-Lipschitz bijection, following Niyogi et al. (2008), with distortion controlled by the reach $\tau$ of $\mathcal{X}$: for a covering resolution $\epsilon \ll \tau$,

$$\|\mathbf{x} - \mathbf{x}'\|_2 \; \leq \; d_{\mathcal{X}}(\mathbf{x}, \mathbf{x}') \; \leq \; \|\mathbf{x} - \mathbf{x}'\|_2 \left(1 + O(\epsilon^2/\tau^2)\right)$$

for $\mathbf{x}, \mathbf{x}' \in \mathcal{X}$ within a ball of coordinate radius $O(\epsilon)$. This ensures that Euclidean distance in $B_d(\mathbf{0}, 1)$ and geodesic distance on $\mathcal{X}$ agree up to a controlled factor, so that all results stated over $B_d(\mathbf{0}, 1)$ in terms of the 2-norm carry over to $\mathcal{X}$ with $D$ replaced by $d$. Hereafter, we adopt the following simplifications: we set $\delta = 0$ (so that $U_{\delta/2}(\mathcal{X}) = \mathcal{X}$, i.e. training and test points lie exactly on the manifold) and use $B_d(\mathbf{0}, 1)$ as our input domain; we take $\{(\mathbf{x}_i, f(\mathbf{x}_i))\}$ with $\mathbf{x}_i \in B_d(\mathbf{0}, 1)$ as our training points; we refer to $f$ as our target function; and we omit boundary effects due to compact support, yielding cleaner results. This allows all subsequent results to be stated directly in terms of the intrinsic dimension $d$, without further reference to the ambient dimension $D$ or the tubular neighborhood $U_{\delta/2}(\mathcal{X})$.

Formally, $\gamma_s$ and $\beta_l$ are defined as follows.

**Definition 1** (Enclosing and Inscribed Radii). *Let $R_i \subseteq B_d(\mathbf{0}, 1)$ denote a decision region induced by the classification network $\mathcal{M}$, and let $B(\mathbf{c}, r) := \{\mathbf{x} \in \mathbb{R}^d : \|\mathbf{x} - \mathbf{c}\|_2 \leq r\}$ denote the closed Euclidean ball of radius $r$ centered at $\mathbf{c}$. Define the* enclosing radius *of $R_i$ as*

$$\gamma(R_i) := \min \{r > 0 \mid \exists \mathbf{c} \in R_i \text{ such that } R_i \subseteq B(\mathbf{c}, r)\},$$

*i.e., the radius of the smallest ball $B(\mathbf{c}, r)$, over all centers $\mathbf{c} \in R_i$, satisfying $R_i \subseteq B(\mathbf{c}, r)$. Define the* inscribed radius *of $R_i$ as*

$$\beta(R_i) := \max \{r > 0 \mid \exists \mathbf{c} \in R_i \text{ such that } B(\mathbf{c}, r) \subseteq R_i\},$$

*i.e., the radius of the largest ball $B(\mathbf{c}, r)$, over all centers $\mathbf{c} \in R_i$, satisfying $B(\mathbf{c}, r) \subseteq R_i$.*

*We then define*

$$\gamma_s := \max_{1 \leq i \leq l} \gamma(R_i), \qquad \beta_l := \min_{1 \leq i \leq l} \beta(R_i),$$

*the largest enclosing radius and smallest inscribed radius, respectively, taken over all decision regions of $\mathcal{M}$.*

**Remark 3.1.** *Under the above simplification, the network $\mathcal{M} : \mathbb{R}^d \to \mathbb{R}^l$, which is trained to learn the target function $f : \mathbb{R}^d \to \mathbb{R}^l$, should be understood as $\mathcal{M} = g \circ \varphi$, where $g$ denotes the network as literally implemented (with input in $\mathbb{R}^D$, or more precisely $U_{\delta/2}(\mathcal{X})$ prior to setting $\delta = 0$) and $\varphi : B_d(\mathbf{0}, 1) \to \mathcal{X}$ is the smooth bijection identifying the intrinsic domain with the data manifold. All covering-number and generalization arguments below are stated for $\mathcal{M}$ and $f$ on $B_d(\mathbf{0}, 1)$; no properties of $g$ beyond those inherited through $\varphi$ are used.*

**Remark 3.2.** *Several lines of prior work address the curse of dimensionality in neural network generalization through structural assumptions on the learning problem. Bach (2017) shows that infinite-width, single-hidden-layer networks trained with convex, variation-norm regularization adapt to unknown* linear *low-dimensional structure in the target function – e.g., dependence only on a low-dimensional projection of the input – so that sample complexity scales with this structural dimension rather than the ambient input dimension. Our approach instead assumes low-dimensional* geometric *structure in the data itself: we take the data to concentrate near a smooth d-dimensional manifold $\mathcal{X} \subset \mathbb{R}^D$ with $d \ll D$, and show (Lemmas 1 and 2) that the sample complexity and generalization error of standard one-hot classification networks — without requiring convex relaxation or unbounded width — scale with the intrinsic dimension d rather than D. These two mechanisms for escaping the curse of dimensionality are complementary: one exploits structure in the target function, the other in the input distribution.*

### 3.2 Generalization bound for covering with fixed radius balls

Our study utilizes specified balls to analyze local geometry, linking this analysis to generalization error and sample complexity. We demonstrate that the parameter $\gamma_s$ effectively characterizes generalization errors. However, a higher sample density is essential to ensure that each decision region contains at least one training point. This necessary density is determined by the maximum radius of the balls, $\beta_l$, within the decision regions.

The lemma below shows that the sample density defined by $\beta_l$ ensures that each region formed by the classification network has at least one training point (proof in Appendix B).

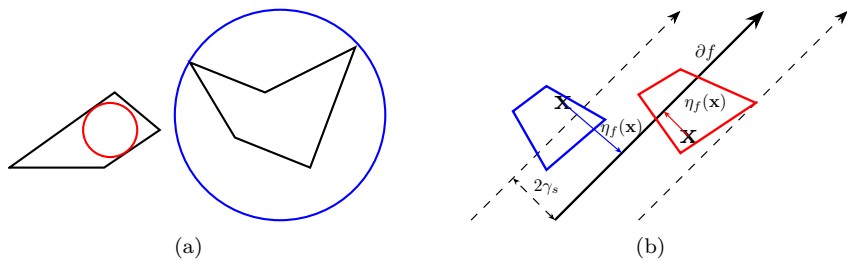

(a)                                                                (b)

Figure 1: (a) Each polygon can be enclosed by its own ball of radius $\gamma_s$ (blue), and contains its own inscribed ball of radius $\beta_l$ (red). (b) Polygons represent decision regions, each assigned a class. Any point $\mathbf{x}$ within the $2\gamma_s$-tube of $\partial f$ (i.e., $\eta_f(\mathbf{x}) \leq 2\gamma_s$) may cause a classification error, contributing to the generalization error bound. The red polygon intersects $\partial f$, with its two subregions belonging to different classes under $f$ but the same class under the network. Polygons farther than $2\gamma_s$ from $\partial f$ cannot intersect the boundary, like the blue polygon.

**Lemma 3.** *Let $\beta_l$ be the radius of the largest inscribed balls in the finest decision regions of the one-hot classification network $\mathcal{M}$, over the input domain $B_d(\mathbf{0}, 1)$. If the training points form a $\beta_l$-covering of $B_d(\mathbf{0}, 1)$ (Assumption (dense coverage in the manifold)), then each decision region of $\mathcal{M}$ contains at least one training point.* ∎

The following lemma shows that, for sufficiently large networks, the generalization error in classification tasks exhibits a linear behavior with respect to $\gamma_s$. The analysis is applicable to classification networks that define piecewise-constant functions over the input domain, with at least one training point assigned to each decision region of a specific class.

Let $f$ be the target classification function. For generalization error, we use the volume of enclosing balls near the actual classification boundary, $\partial f$, based on the observation that when $\partial f$ intersects with the network's decision regions, potential errors arise. Specifically, points within $2\gamma_s$ of $\partial f$ are likely to belong to a region that could contribute to errors. This relationship emphasizes $\gamma_s$'s significance in determining the network's generalization error, as shown in Figure 1(b). In addition, to measure the volume of balls proximity to $\partial f$, denoted by $2\gamma_s$, we need to characterize the boundary over the input domain. Here, we assume $\partial f$ is a semi-algebraic set because it is tame and excludes infinite oscillation and fractal phenomena. A semi-algebraic set has arbitrarily high finite algebraic complexity, but not arbitrarily pathological boundaries.

The process of stratification involves decomposing a semi-algebraic set into a finite union of disjoint semi-algebraic subsets, known as strata, which are locally closed submanifolds (Benedetti et al., 1991). In a stratification, the boundary of the entire set $S$ is the union of all strata of lower dimension than the "top-level" interior strata. If $S$ is a closed semi-algebraic set, then $\partial S = \bigcup_{X_j \cap \text{int}(S) = \emptyset} X_j$, where $X_j$ is a stratum of $S$ and it does not touch the interior of $S$. We use the notation $|S|$ to denote the volume of the set $S$.

The target function $f$ partitions the input domain into a finite union of disjoint sets, denoted as $\bigcup c(i)$ where $i \in [l]$. The set $c(i) = \{\mathbf{x} \in B_d(\mathbf{0}, 1) | f(\mathbf{x}) = \mathbf{e}_i\}$ corresponds to class $i$ and is assumed to be a semi-algebraic set, meaning that its representation is a finite union of disjoint subsets, each defined by a finite combination of polynomial equations and inequalities. Formally, we can represent a stratification of $\bigcup c(i)$ by a finite collection of semi-algebraic subsets $\{f_k = \bigcup_j f_k^j\}$, where the dimension of $f_k^j$ is $k$ and $f_k^j$ are disjoint semi-algebraic sets. We refer to the volume of $f_k^j$ as $|f_k^j|$. The concept of volume measures the size of an object: a 1-volume corresponds to length, while a 2-volume corresponds to area, and so on.

Hence, considering the boundary of the involved sets leads to the following relationship:

$$\partial f = \bigcup_{i=1}^{l} \partial c(i) = \bigcup_{k=0}^{d-1} f_k = \bigcup_{k=0}^{d-1} \bigcup_{j} f_k^j, \tag{8}$$

with $f_k = \bigcup_j f_k^j$ and

$$|f_k| = \sum_j |f_k^j|. \tag{9}$$

For example, consider the stratification of $c(1) = \{(x, y) \in \mathbb{R}^2 \mid |x| + |y| < 1\} \bigcup \{(x, y) \in \mathbb{R}^2 \mid |x| + |y| = 1\}$ with dimension 2. The first subset of that stratification is the interior of $g(1)$; therefore, that set does not contribute to the boundary of $g(1)$. We can decompose the second subset into four 1-dimensional semi-algebraic sets: $f_1^1 = \{x + y = 1 | x, y > 0\}$, $f_1^2 = \{-x + y = 1 | x < 0, y > 0\}$, $f_1^3 = \{-x - y = 1 | x, y < 0\}$, and $f_1^4 = \{x - y = 1 | x > 0, y < 0\}$, and four 0-dimensional vertices: $f_0^1 = \{(1, 0)\}$, $f_0^2 = \{(0, 1)\}$, $f_0^3 = \{(-1, 0)\}$, and $f_0^4 = \{(0, -1)\}$. The boundaries of their stratification of $c(1)$ are as follows:

$$\partial c(1) = f_1 \cup f_0 = (\cup_i f_1^i) \cup (\cup_j f_0^j). \tag{10}$$

The following lemma demonstrates that assuming the decision boundary is a semi-algebraic set leads to its decomposition into a combination of polynomial functions of varying degrees (proof in Appendix C). This lemma connects the generalization error to the network's granularity, represented by $\gamma_s$ and $\beta_l$, in approximating the classification boundary. For networks with a small $\gamma_s$, the decision boundary that intersects a given region can be considered a curved $(d-1)$-dimensional surface, homeomorphic to a hyperplane. The $2\gamma_s$-neighborhood of the union of the "hyperplanes" represents the region in the input domain that can contribute to a bound of generalization error.

**Lemma 4.** *Let $f : B_d(\mathbf{0}, 1) \to \{\mathbf{e}_1, \cdots, \mathbf{e}_l\}$ denote the target function, and let $c(i) := f^{-1}(\mathbf{e}_i) = \{\mathbf{x} \mid f(\mathbf{x}) = \mathbf{e}_i\}$ denote the i-th class region, which we assume is a semi-algebraic subset of $B_d(\mathbf{0}, 1)$, so that the domain is partitioned as $B_d(\mathbf{0}, 1) = \bigcup_{i=1}^l c(i)$, with decision boundary $\partial f$ adhering to the stratification equations (8) and (9). Let $\gamma_s$ and $\beta_l$ denote the smallest enclosing radius and largest inscribed radius of the decision regions in $B_d(\mathbf{0}, 1)$ of the one-hot classification network $\mathcal{M}$, and let $\gamma_0$ denote the critical radius of $\partial f$ (Definition 4). Assume $\gamma_s \leq \gamma_0/2$, the training data $\beta_l$-covers the unit d-ball (Lemma 3), and the data satisfies Assumption (histogram approximation or interpolation). We let $\mathbf{1}_{\text{hist}} \in \{0, 1\}$ denote the condition indicator, where $\mathbf{1}_{\text{hist}} = 1$ corresponds to condition (i) (histogram approximation) and $\mathbf{1}_{\text{hist}} = 0$ to condition (ii) (interpolation).*

*(i) For an arbitrary density function $\mathcal{P}$ on $B_d(\mathbf{0}, 1)$, the error is given by:*

$$\int_{B_d(\mathbf{0},1)} g(f(\mathbf{x}), \mathcal{M}(\mathbf{x})) \mathcal{P}(\mathbf{x}) \, d\mathbf{x} \leq \mathbf{1}_{\text{hist}} \frac{1}{N} \sum_{i=1}^N g(f(\mathbf{x}_i), \mathcal{M}(\mathbf{x}_i))$$
$$+ c_{\mathcal{P}} \sum_{k=0}^{d-1} \left[ C_0(\gamma_s) \cdot (2\gamma_s)^{d-k} |B_{d-k}(\mathbf{0}, 1)| |f_k| + \tfrac{1}{2} (2\gamma_s)^{d-k+1} |B_{d-k+1}(\mathbf{0}, 1)| |\partial f_k| \right], \tag{11}$$

*where $g(f(\mathbf{x}), \mathcal{M}(\mathbf{x}))$ is the 0/1 error, $C_0(\gamma_s)$ is the uniform curvature-correction factor of Lemma 5 (with $C_0(\gamma_s) \to 1$ as $\gamma_s \to 0$), $|\partial f_k| := \sum_i |\partial f_k^i|$ denotes the total $(k-1)$-dimensional frontier volume of stratum $f_k$, with $\partial f_k^i$ and $|\partial f_k^i|$ as defined in the proof of Lemma 5, and $c_{\mathcal{P}} \in [k, K]$, with $k$ and $K$ as defined in Assumption (bounded support), denotes the average density of $\mathcal{P}$ over the $2\gamma_s$-neighborhood of $\partial f$.*

*(ii) We further suppose the volume of $f_{d-1}$ is not zero. For small values of $\gamma_s$, the frontier correction terms in (11) are of strictly higher order in $\gamma_s$ than the leading term $(2\gamma_s) |B_1(\mathbf{0}, 1)| |f_{d-1}|$, and the curvature-correction factor satisfies $C_0(\gamma_s) \to 1$ as $\gamma_s \to 0$; we treat this factor as approximately 1 to leading order, though we do not establish its precise rate of convergence here. Under this leading-order approximation, we can approximate the above expression as follows:*

$$\int_{B_d(\mathbf{0},1)} g(f(\mathbf{x}), \mathcal{M}(\mathbf{x})) \mathcal{P}(\mathbf{x}) \, d\mathbf{x} \leq \mathbf{1}_{\text{hist}} \frac{1}{N} \sum_{i=1}^N g(f(\mathbf{x}_i), \mathcal{M}(\mathbf{x}_i)) + c_{\mathcal{P}} (4\gamma_s) |f_{d-1}|. \tag{12}$$

∎

## 4 Connecting geometry to scaling laws

### 4.1 Box-model dimension and scaling laws with respect to network size

Having bounded the generalization error in terms of ball radii capturing the local geometry of a network function, we now investigate how these radii relate to the number of network parameters.

#### 4.1.1 Box-model dimension

The box-counting dimension is a way to measure the "ruggedness" or complexity of a shape by examining how it occupies space at different scales. To calculate the box-counting dimension of a specific shape, we examine the relationship between the scaling factor and the number of boxes needed to cover it. The dimension $D$ is the logarithmic power law relationship between the number of boxes and their size:

$$D = \lim_{s \to 0} \frac{\log \#B(s)}{\log(1/s)}, \tag{13}$$

where $\#B$ is the number of boxes (a grid of squares) with side length $s$ over the object. This dimension is found by overlaying a grid of boxes with side length $s$ over the object, counting the number of boxes, then making the boxes smaller and counting again, and observing how the count increases as $s$ decreases. In practice, since the box-counting dimension may not exist or may be numerically difficult to calculate, the upper box-counting dimension is used to indicate that this set has a dimension at least $\bar{D}$ by considering $\bar{D} = \limsup_{s \to 0} \frac{\log \#B(s)}{\log(1/s)}$.

The box-counting dimension measures the complexity of a set by quantifying how many boxes are needed to describe its shape at various scales of box size. In contrast, the box-model dimension measures the number of parameters $\#W$ in a model $\mathcal{M}$ required to construct the box-coverage of a target shape when one zooms into the shape. We define the box-model dimension as follows:

$$D_{\mathcal{M}} = \lim_{s \to 0} \frac{\log \#W(s)}{\log(1/s)}. \tag{14}$$

By this definition, we focus on how the radii decrease with depth as the number of parameters in a neural network increases. We express the relationship between the number of parameters $\#W$ in network $\mathcal{M}$ and the radius $\gamma_s$ as follows:

$$\gamma_s \leq \frac{1}{\#W^{1/\log_2(1+\alpha(\#W, \gamma_s))}}. \tag{15}$$

In our analysis of classification tasks, the radius $\gamma_s$ is derived from the decision regions of model $\mathcal{M}$. Increasing the number of network parameters $\#W$ generates more decision regions, potentially reducing the radius. As a result, $\gamma_s$ is a non-increasing function of $\#W$. The condition in Eq. (15) states that the growth rate of $\#W^{1/\log_2(1+\alpha)}$ must be slower than the decrease in $\gamma_s$, as characterized by the value of $\alpha$. This condition provides an upper bound of parameters, preventing the network from overfitting or over-approximating the training points. Taking the logarithm on both sides and then dividing by $\log_2 1/\gamma_s$, we obtain:

$$\frac{\log_2 \#W}{\log_2 1/\gamma_s} \leq \log_2(1 + \alpha(\#W, \gamma_s)). \tag{16}$$

As $\gamma_s \to 0$, we obtain the upper box-model dimension:

$$\limsup_{\gamma_s \to 0} \frac{\log_2 \#W}{\log_2 1/\gamma_s} = \bar{D}_{\mathcal{M}} \to \log_2(1 + \alpha_{\mathcal{M}}). \tag{17}$$

Here, the parameter $\alpha_{\mathcal{M}} > 0$ is model-dependent and influenced by factors such as the network architecture and the specific training method used. Due to this model dependence, networks with the same input domain can produce different box-model dimensions depending on their architecture and training method.

In Appendix D, we demonstrate that the exact configuration of the input domain, generated by shallow and deep network structures, can produce different box-model dimensions in the scaling laws.

The logarithmic function in Eq. (15) arises from constructing a base network $\mathcal{M}_0$ and expanding its depth to create $\mathcal{M}_1$. The goal was to ensure that $\gamma_1$ of $\mathcal{M}_1$ was reduced to less than half the value of $\gamma_0$ of $\mathcal{M}_0$, while increasing the parameter count by a factor of at most $1 + \alpha$, where $\gamma_0$ and $\alpha$ are held constant throughout the construction. We then apply the process recursively to decrease the enclosing radius to a desired value $\gamma_s$. Setting $\gamma_0 = 1$, reducing the radius to $\gamma_s$ requires $L = -\log_2 \gamma_s$ iterations. Let the resulting network be $\mathcal{M}_L$. Using $\#W_{l_k} \leq (1 + \alpha)\#W_{l_{k-1}}$ with initial condition $\#W_0 = 1$, the parameter count of $\mathcal{M}_L$ satisfies $\#W \leq (1 + \alpha)^L$, and hence

$$\log_2 \#W \leq (-\log_2(1 + \alpha))(\log_2 \gamma_s). \tag{18}$$

This gives the relation $\gamma_s \cdot \#W^{\frac{1}{\log_2(1+\alpha)}} \leq 1$ with $\alpha > 0$, as depicted in Eq. (15). We note that $\alpha$ need not be uniform across sub-networks or depths; the construction readily accommodates depth-varying $\alpha$.

We consider the network $\mathcal{M}_L^0 = [I + \rho M_L] \circ [I + \rho M_{L-1}] \circ \cdots \circ [I + \rho M_1]$, where $I$ denotes the identity mapping (serving as a skip connection within each layer), $\rho$ denotes the Rectified Linear Unit (ReLU) activation function, and $M_l$ represents an affine linear mapping. We show that it is possible to arrange the hyperplanes and the training points such that the scaling law is satisfied with a constant exponent for any depth $l$ in $\mathcal{M}_l^0$. The arrangement of the hyperplanes and training points required here is not generally achievable by gradient-based training methods. Furthermore, our construction halves the covering radius at each successive layer. Since the number of covering balls scales exponentially with each such reduction, this halving condition leads to an exponential increase in network width from one layer to the next. This contrasts with practical architectures such as ResNets and VGGs, where network width grows only moderately (e.g., polynomially) across layers. Therefore, the exponential parameter requirement is a property of our theoretical construction, introduced to establish the existence of scaling laws, and should not be interpreted as a constraint on practical architectures. A detailed derivation of this proposition can be found in Appendix E.

**Proposition 1.** *Consider networks with input domain $B_d(\mathbf{0}, 1)$ (recall $d$ is the intrinsic dimension of $\mathcal{X}$, per the Simplification of Notation in Section 3.1). Let $\#W_l$ denote the number of parameters and $\gamma_l$ denote the radius of the smallest enclosing balls of the regions induced by $\mathcal{P}_l^0$ in network $\mathcal{M}_l^0$. There exists a configuration of hyperplanes in layer $\rho M_{l+1}$ and a choice of training points such that the finest partition $\mathcal{P}_{l+1}^0$ of network $\mathcal{M}_{l+1}^0$ with $l \leq L - 1$ is a refinement of $\mathcal{P}_l^0$ that satisfies $\gamma_{l+1} \leq \frac{\gamma_l}{2}$. Additionally, the number of parameters in $\mathcal{M}_{l+1}$ and $\mathcal{M}_l$ can be constrained by the inequality $\#W_{l+1} \leq (1 + \alpha)\#W_l$, where $\alpha \geq 1$ when $l$ is large (as the covering radius becomes smaller).* ■

The construction in this proposition focuses on how to achieve the desired covering radius by increasing network width at each layer. Once the desired covering radius of $\mathcal{M}_L^0$ is achieved, we align the output dimension with that of the target network by appending a series of layers of gradually decreasing width. This width-decreasing composition preserves the covering radius achieved by $\mathcal{M}_L^0$.

A primary property of the constructed network, guaranteed by Proposition 1, is given by the formula:

$$\#W_{l+1}(\gamma_{l+1}) \leq (1 + \alpha)\#W_l(\gamma_l), \tag{19}$$

which leads to $\gamma_{l+1} \leq \frac{\gamma_l}{2}$. In Subsection 5, we shall show that this property satisfies discrete scale invariance and consequently leads to a log-log linear scaling law with a slope $\frac{1}{\log_2(1+\alpha)}$ when $l$ is large. In Lemma 4, we have connected $\gamma_l$ to the generalization error bound of network $\mathcal{M}_l^0$.

The connection between the reduction of $\gamma_l$ by a factor strictly smaller than 1 and the increase of $\#W_l$ by a factor larger than 1 analytically demonstrates that the scaling law depends on both the architecture design and the learning method, where the former controls the number of parameters and the latter binds the parameters to the generalization error bound.

### 4.1.2  Scaling laws with respect to network size

We now examine the implications of the condition in Eq. (15) for generalization bounds. Given $\gamma_s$ and $\#W$, the value of $\alpha$ satisfying the condition has a minimal value. We denote by $\alpha^*(\#W, \gamma_s)$ the minimal

value that satisfies the condition with equality. By substituting this equality condition into Lemma 4 for $\gamma_s$, the generalization bound becomes inversely proportional (up to a logarithmic exponent) to the number of parameters in the network, as made precise in Theorem 1.

**Theorem 1.** *Suppose the classification boundary $\partial f$ of the target classification function $f : B_d(\mathbf{0}, 1) \to \{\mathbf{e}_1, \cdots, \mathbf{e}_l\}$ is a semi-algebraic set satisfying the stratification equations (8) and (9). Let $\gamma_s$ and $\beta_l$ denote the smallest enclosing radius and the largest inscribed radius of the partition regions of $B_d(\mathbf{0}, 1)$ induced by a one-hot classification network $\mathcal{M} : B_d(\mathbf{0}, 1) \to \{\mathbf{e}_1, \cdots, \mathbf{e}_l\}$ with parameter size $\#W$. Let $\alpha^*(\#W, \gamma_s)$ denote the value satisfying*

$$\gamma_s = \#W^{-1/\log_2(1+\alpha^*(\#W,\gamma_s))}. \tag{20}$$

*Assume $\gamma_s \leq \gamma_0/2$, where $\gamma_0$ is the critical radius of $\partial f$ (Definition 4), the training data $\beta_l$-covers the unit $d$-ball and satisfies Assumption (histogram approximation or interpolation), and $\mathcal{P}$ satisfies Assumption (bounded support). We let $\mathbf{1}_{\text{hist}} \in \{0, 1\}$ denote the condition indicator, where $\mathbf{1}_{\text{hist}} = 1$ corresponds to condition (i) (histogram approximation) and $\mathbf{1}_{\text{hist}} = 0$ to condition (ii) (interpolation).*

*(i) Substituting the relation $\gamma_s = \#W^{-1/\log_2(1+\alpha^*(\#W,\gamma_s))}$ into Lemma 4(i), we bound the generalization error of $\mathcal{M}$ with respect to $f$ as follows:*

$$\int_{B_d(\mathbf{0},1)} g(f(\mathbf{x}), \mathcal{M}(\mathbf{x}))\mathcal{P}(\mathbf{x}) \, d\mathbf{x} \leq \mathbf{1}_{\text{hist}} \frac{1}{N} \sum_{i=1}^{N} g(f(\mathbf{x}_i), \mathcal{M}(\mathbf{x}_i))$$

$$+ c_{\mathcal{P}} \sum_{k=0}^{d-1} \left[ C_0(\gamma_s) \cdot \left( \frac{2}{\#W^{1/\log_2(1+\alpha^*(\#W,\gamma_s))}} \right)^{d-k} |B_{d-k}(\mathbf{0}, 1)| |f_k| \right.$$

$$\left. + \tfrac{1}{2} \left( \frac{2}{\#W^{1/\log_2(1+\alpha^*(\#W,\gamma_s))}} \right)^{d-k+1} |B_{d-k+1}(\mathbf{0}, 1)| |\partial f_k| \right], \tag{21}$$

*where $g(f(\mathbf{x}), \mathcal{M}(\mathbf{x}))$ is the 0/1 error, $C_0(\gamma_s)$ is the uniform curvature-correction factor of Lemma 5 (with $C_0(\gamma_s) \to 1$ as $\gamma_s \to 0$), $|\partial f_k|$ is as defined in Lemma 4, and $c_{\mathcal{P}} \in [k, K]$, with $k$ and $K$ as defined in Assumption (bounded support), depends on the density $\mathcal{P}$ restricted to the $2\gamma_s$-neighborhood of $\partial f$.*

*(ii) We further suppose the volume of $f_{d-1}$ is not zero. For a small value of $\gamma_s$, the frontier correction terms in (21) are of strictly higher order in $\gamma_s$ than the leading term, and the curvature-correction factor satisfies $C_0(\gamma_s) \to 1$; treating $C_0(\gamma_s)$ as approximately 1 to leading order, we can approximate the above expression as follows:*

$$\int_{B_d(\mathbf{0},1)} g(f(\mathbf{x}), \mathcal{M}(\mathbf{x}))\mathcal{P}(\mathbf{x}) \, d\mathbf{x} \leq \mathbf{1}_{\text{hist}} \frac{1}{N} \sum_{i=1}^{N} g(f(\mathbf{x}_i), \mathcal{M}(\mathbf{x}_i)) + \frac{4 c_{\mathcal{P}} |f_{d-1}|}{\#W^{1/\log_2(1+\alpha^*(\#W,\gamma_s))}}. \tag{22}$$

*(iii) Moreover, as $\#W \to \infty$ (equivalently, $\gamma_s \to 0$), the exponent $\alpha^*(\#W, \gamma_s)$ converges to the model-family-specific constant $\alpha_{\mathcal{M}}$, where $\bar{D}_{\mathcal{M}} = \log_2(1 + \alpha_{\mathcal{M}})$ is the box-model dimension defined in Eq. (17). Consequently, the bound in (ii) converges to, and is asymptotically upper bounded by,*

$$\int_{B_d(\mathbf{0},1)} g(f(\mathbf{x}), \mathcal{M}(\mathbf{x}))\mathcal{P}(\mathbf{x}) \, d\mathbf{x} \leq \mathbf{1}_{\text{hist}} \frac{1}{N} \sum_{i=1}^{N} g(f(\mathbf{x}_i), \mathcal{M}(\mathbf{x}_i)) + \frac{4 c_{\mathcal{P}} |f_{d-1}|}{\#W^{1/\bar{D}_{\mathcal{M}}}}, \tag{23}$$

*for sufficiently large $\#W$ (equivalently, sufficiently small $\gamma_s$).* ∎

*Proof.* (i) follows directly by substituting $\gamma_s = \#W^{-1/\log_2(1+\alpha^*(\#W,\gamma_s))}$ into Lemma 4(i), retaining both the curvature-correction factor $C_0(\gamma_s)$ and the frontier terms as they appear there. (ii) follows by the same substitution into Lemma 4(ii), where, as in the lemma, the frontier terms are dropped as strictly higher order in $\gamma_s$, and $C_0(\gamma_s)$ is treated as approximately 1 to leading order.

(iii) Taking $\log_2$ of both sides of $\gamma_s = \#W^{-1/\log_2(1+\alpha^*(\#W,\gamma_s))}$ gives

$$\log_2(1 + \alpha^*(\#W, \gamma_s)) = \frac{\log_2 \#W}{\log_2(1/\gamma_s)}. \tag{24}$$

By the definition of the box-model dimension (17),

$$\limsup_{\gamma_s \to 0} \log_2(1 + \alpha^*(\#W, \gamma_s)) = \limsup_{\gamma_s \to 0} \frac{\log_2 \#W}{\log_2(1/\gamma_s)} = \bar{D}_{\mathcal{M}} = \log_2(1 + \alpha_{\mathcal{M}}). \tag{25}$$

Fix $\epsilon > 0$. By definition of $\limsup$, there exists $\delta_\epsilon > 0$ such that for all $\gamma_s < \delta_\epsilon$,

$$\log_2(1 + \alpha^*(\#W, \gamma_s)) \leq \log_2(1 + \alpha_{\mathcal{M}}) + \epsilon. \tag{26}$$

Therefore, for all $\gamma_s < \delta_\epsilon$,

$$\frac{1}{\#W^{1/\log_2(1+\alpha^*(\#W,\gamma_s))}} \leq \frac{1}{\#W^{1/(\log_2(1+\alpha_{\mathcal{M}})+\epsilon)}}, \tag{27}$$

and since this holds for every $\epsilon > 0$ (with $\delta_\epsilon$ depending on $\epsilon$), letting $\epsilon \to 0$, we obtain the asymptotic bound

$$\frac{1}{\#W^{1/\log_2(1+\alpha^*(\#W,\gamma_s))}} \lesssim \frac{1}{\#W^{1/\log_2(1+\alpha_{\mathcal{M}})}} \tag{28}$$

as $\gamma_s \to 0$. Substituting into (22) yields (23).

$\square$

In this theorem, $\#W$ denotes the number of parameters and $\gamma_s$ is the radius of the smallest enclosing ball among the decision regions on $B_d(\mathbf{0}, 1)$ induced by the network $\mathcal{M}$. Fix $\#W$: different learning algorithms can yield different decision partitions and their geometric parameters $\gamma_s$, and hence different values of $\alpha^*(\#W, \gamma_s)$ (in Section 6.1.1, the VGG models trained with and without batch normalization on ImageNet yield different estimated box-model dimensions). On the other hand, networks that produce the same value of $\gamma_s$ can have different architectures (in Appendix D, our network and the network of Sharma and Kaplan (2022) attain the same $\gamma_s$ but with a different number of parameters). Thus, the value of $\alpha$ depends on both the learning algorithm and the network architecture. The above theorem shows that, asymptotically, a scaling law exists and can be characterized by a box-model dimension in the log-log plane. In practice, however, the box-model dimension must be estimated by fitting the observed curve of $\alpha^*$ for large networks, rather than computed directly, as discussed in Section 6.

For small networks in the regime of part (i), the generalization error bound is complicated because it is a summation of several terms inversely proportional to the network size: $c_{\mathcal{P}} \sum_{k=0}^{d-1} \left( \frac{2}{\#W^{1/\log_2(1+\alpha^*(\#W,\gamma_s))}} \right)^{d-k} |B_{d-k}(\mathbf{0},1)| \|f_k\|$. In contrast, for sufficiently large networks in the regime of parts (ii) and (iii), the relation between the generalization error bound and the network size admits a simpler characterization.

We denote the (upper bound on the) generalization error as a function of $(\#W, \gamma_s)$ by $g(\#W, \gamma_s) := \frac{4c_{\mathcal{P}}|f_{d-1}|}{\#W^{1/\log_2(1+\alpha^*(\#W,\gamma_s))}}$, as given by Theorem 1(ii). Taking the logarithm of both sides, we obtain

$$\ln g(\#W, \gamma_s) + \frac{1}{\log_2(1 + \alpha^*(\#W, \gamma_s))} \ln \#W = k, \tag{29}$$

for the constant $k := \ln(4c_{\mathcal{P}}|f_{d-1}|)$. This curve relates the generalization error bound to the network size, with its complexity governed by $\log_2(1+\alpha^*(\#W, \gamma_s))$. In this regime, the coefficient $1/\log_2(1+\alpha^*(\#W, \gamma_s))$ varies with both $\#W$ and $\gamma_s$, so the scaling law need not be log-log-linear.

As $\#W \to \infty$, entering the regime of part (iii), Theorem 1(iii) gives $\alpha^*(\#W, \gamma_s) \to \alpha_{\mathcal{M}}$, and Eq. (29) reduces to a log-log-linear curve with fixed slope $\log_2(1 + \alpha_{\mathcal{M}})^{-1}$, characterizing the asymptotic scaling law of the

generalization error for sufficiently large networks. This log-log-linear form applies across model families; the specific slope $\log_2(1+\alpha_{\mathcal{M}})^{-1}$, governed by the box-model dimension, varies by model family. This slope serves as an index of how efficiently a network family reduces generalization error as its parameter count increases.

Theorem 1 indicates that the log-log-linear scaling law of part (iii) is an asymptotic relationship, valid once the network is sufficiently large (equivalently, $\gamma_s$ sufficiently small) that $\alpha^*(\#W, \gamma_s)$ has converged to the model-family constant $\alpha_{\mathcal{M}}$. For smaller networks, part (i) shows the generalization error bound is instead a sum of several $\#W$-dependent terms with no single fixed slope, so the asymptotic law fit on larger models is not expected to extrapolate backward to accurately predict the performance of substantially smaller networks. This decomposition parallels recent theoretical work in linear regression (Lin et al., 2024) and solvable random feature models (Maloney et al., 2022), both of which show that the classical bias-variance decomposition and the empirically-observed monotone neural scaling law are reconciled only in specific asymptotic regimes, rather than holding uniformly across all model sizes.

A separate line of work explains neural scaling laws through training dynamics rather than model capacity. Ren et al. (2025) introduce a hierarchical multi-index target, a sum of single-index sub-tasks with power-law decaying weights $k^{-\gamma}$, and study how SGD learns this hierarchy over time. Defilippis et al. (2026) sharpen this into exact information-theoretic scaling laws, revealing plateaus and phase transitions as each sub-task is sequentially resolved. These results derive power-law scaling from the difficulty structure of the target function itself. In contrast, our framework derives scaling laws from the geometry of the decision boundary and the covering complexity of the data manifold, independent of any specific target hierarchy. The two approaches address different sources of scaling behavior and are, in this sense, complementary.

**Predicting larger models from smaller ones.** The regime of part (iii) fits the conventional log-log-linear scaling law, and it is in this regime that predicting the performance of larger models from smaller ones can be reliably applied. However, since it is not easy to determine how small $\gamma_s$ must be for part (iii) to apply, the prediction problem on the $(\#W, g)$ plane is nontrivial: if the sampled (smaller) models have not yet entered the asymptotic regime, the fitted curve may reflect the more complex, non-asymptotic behavior of part (i) or the transition between (i) and (iii), rather than the true limiting slope $\log_2(1+\alpha_{\mathcal{M}})$, and the resulting extrapolation to larger models can be unreliable. As reported by Caballero et al. (2022), this kind of extrapolation breakdown is indeed observed in practice.

## 5 Discrete Scale Invariance

From the perspective of both software and hardware, deep learning is built from modular design blocks, such as standard transformer blocks or ResNet blocks. When we scale a model, we do not arbitrarily add individual parameters throughout the computation graph. Instead, we stack these self-contained modules. This structural modularity provides a natural discretization of the network's capacity. Each additional block contributes a discrete unit of function-fitting capability, and their recursive stacking gives rise to discrete self-similarity and scale invariance. The following analysis shows that the generalization error bound for a model family constructed via this modular design exhibits approximately linear behavior in log-log plots of error versus parameter count, suggesting the potential for simple extrapolation from a small set of models to estimate the performance of larger models. From a practical design perspective, this could be useful because, before a model is scaled, such an extrapolation may help inform whether scaling is likely to be worthwhile, and provide a rough estimate of the potential performance gain, potentially reducing the need for extensive hyperparameter tuning at each scale.

The modular-based design can be related to discrete scale invariance. Sornette (1998) investigate discrete scale invariance, the behavior of $f(\lambda \mathbf{x}) = \lambda^p f(\mathbf{x})$ for some discrete $\lambda$. The general solution to this equation is not a pure power law. Instead, it is the power law modified by a periodic function: $f(\mathbf{x}) = \mathbf{x}^p F\left(\frac{\ln \mathbf{x}}{\ln \lambda}\right)$, where $F(\mathbf{y})$ is a periodic function of period 1 (meaning that $F(\mathbf{y} + 1) = F(\mathbf{y})$). Since $F$ is positive and bounded, taking the logarithmic slope yields

$$\frac{\ln f(\mathbf{x})}{\ln 1/\mathbf{x}} = -p + \frac{\ln F\left(\frac{\ln \mathbf{x}}{\ln \lambda}\right)}{\ln 1/\mathbf{x}}, \tag{30}$$

which shows that discrete scale invariance manifests as log-periodic oscillations superimposed on a leading power law. Taking the limit $\mathbf{x} \to 0$, the oscillatory term vanishes since $F$ is positive and bounded, giving

$$\lim_{\mathbf{x} \to 0} \frac{\ln f(\mathbf{x})}{\ln 1/\mathbf{x}} = -p. \tag{31}$$

The network constructed in Proposition 1 satisfies the discrete scale invariance equation: $\#W(\gamma/2) = (1 + \alpha)\#W(\gamma)$ with $\lambda = 1/2$ and $p = -\log_2(1 + \alpha)$. Then, $\lim_{\gamma \to 0} \frac{\ln \#W(\gamma)}{\ln 1/\gamma} = -p = \log_2(1 + \alpha)$, the box-model dimension.

Since $\gamma$ is related to the generalization error bound (Lemma 4), we can apply the framework of discrete scale invariance to show that the log-log plot of generalization error against parameter count forms a straight line modulated by a highly regular oscillation. This linear relationship allows extrapolation in either direction along the parameter axis: the test performance of larger models can be predicted from that of smaller ones, and conversely, the performance of smaller models can be inferred from larger ones.

Following this perspective, let $\#W(\gamma)$ represent the number of parameters in a network, where $\gamma$ denotes the smallest enclosing radius of its partition of the input domain. Suppose we construct a larger network by adding $\alpha\#W(\gamma)$ parameters to the original network while simultaneously reducing the smallest enclosing radius to $\lambda_0\gamma$, where $\lambda_0 < 1$ is a constant. We can express this construction using the following equation for discrete scale invariance:

$$\#W(\lambda_0\gamma) = (1 + \alpha)\#W(\gamma) = \lambda_0^{\frac{\log_2(1+\alpha)}{\log_2 \lambda_0}}\#W(\gamma), \tag{32}$$

which identifies $p = \frac{\log_2(1+\alpha)}{\log_2 \lambda_0}$ in the general framework.

Hence, the solution to the above equation is

$$\frac{\ln \#W(\gamma)}{\ln 1/\gamma} = -\frac{\log_2(1 + \alpha)}{\log_2 \lambda_0} + \frac{\ln F\left(\frac{\ln \gamma}{\ln \lambda_0}\right)}{\ln 1/\gamma}. \tag{33}$$

Taking the limit $\gamma \to 0$, we define the box-model dimension as

$$\lim_{\gamma \to 0} \frac{\ln \#W(\gamma)}{\ln 1/\gamma} = -\frac{\log_2(1 + \alpha)}{\log_2 \lambda_0} = \log_2(1 + \alpha_{\mathcal{M}}). \tag{34}$$

## 6 Limitations and Future Work

Our analysis relies on several simplifying assumptions that suggest natural directions for future work. First, our geometric arguments assume the classification boundary is semi-algebraic with a well-defined, strictly positive reach, which excludes decision boundaries with more pathological local geometry; extending the framework to a broader class of boundaries, or relaxing the reach condition to hold only approximately, would widen the applicability of our bounds. Second, our density assumptions on $\mathcal{P}$ (bounded above and below on its support) are convenient for deriving clean bounds but may not hold for all data distributions of practical interest; understanding how the bounds degrade under weaker density assumptions remains open.

Assumption (histogram approximation and interpolation) defines two conditions, either of which being satisfied is sufficient for the scaling law depicted by Theorem 1 to hold. Even though, in theory, the histogram approximation holds because $\frac{n_p}{N}$, which estimates $\int_p \mathcal{P}(\mathbf{x})\,d\mathbf{x}$, is an unbiased estimator with variance converging to 0 under our density assumption on $\mathcal{P}$ as $N \to \infty$, the histogram approximation is hard to evaluate at finite $N$. This is because the assumption must hold uniformly across all decision regions, and both the number of decision regions and the boundary of each decision region are computationally hard to delineate for a deep neural network classifier (Serra et al., 2018; Stargalla et al., 2025).

In contrast, the interpolation condition can be evaluated. In our analysis, the interpolation condition corresponds to zero training error. In practice, this condition can be approximated with small training error.

Below in Subsection 6.1, we provide illustrative examples showing that deep networks built from modular designs and trained by minimizing empirical risk generate a linear scaling law in the log-log plot. Even using simple prediction methods (Method A and Method B in Subsection 6.1), most predictions from smaller models to larger models, and from larger models to smaller models, yield acceptable results. However, some predictions require further scrutiny, such as the prediction of ViT-22B from ViT-g/14 and ViT-G/14, which yields a 22% relative error (the last table in Appendix F). Since we rely on publicly available datasets and their benchmark reports, our illustrations alone are not exhaustive enough to fully support our analysis. Publicly available performance reports do not report the training error. Thus, we cannot directly derive the generalization error from these reports. However, judging by the number of parameters adopted compared to the size of the training data, we can assume that they are close to the interpolation zone. This is because empirical risk minimization algorithms typically converge close to the interpolation zone in the over-parameterized case. Thus, using the data in these reports to verify our analysis is a reasonable, if imperfect, approximation.

We do not evaluate geometric parameters such as $\gamma_s$ and $\beta_l$ for a trained network to verify our theoretical results directly. These quantities are defined over the network's decision boundary, and, as we explain below, are difficult to compute in practice. A deep neural network with piecewise-linear activations (e.g., ReLU) can induce a number of decision regions that grows exponentially in depth, as shown by Montufar et al. (2014) and tightened by Serra et al. (2018).

Given the large number of decision regions a network can induce, delineating the boundary of each decision region is itself a difficult computational and theoretical problem, so these values remain difficult to compute directly. Further, $\gamma_s$ and $\beta_l$ depend on the geometry of the underlying data manifold, which itself is generally unknown and must be estimated from finite samples, compounding the difficulty. Altogether, we do not currently have an effective method to evaluate their values empirically in this paper; our empirical validation instead focuses on the overall log-log linear trend between generalization error and parameter count, rather than direct measurement of these geometric quantities. Developing tractable estimators or proxies for $\gamma_s$ and $\beta_l$, for instance by approximating the reach or boundary curvature from finite samples, is an important direction for connecting our geometric bounds more directly to empirical measurement.

The value of $\alpha^*$ is defined as $\gamma_s \cdot \#W^{\frac{1}{\log_2(1+\alpha^*(\#W, \gamma_s))}} = 1$. Because $\gamma_s$ is hard to obtain, this value of $\alpha$ cannot be directly obtained. We thus estimate this value by comparing two points, as demonstrated in Subsection 6.1, using the fact that $\gamma_s$ is proportional to the generalization error, which is measurable when test and training performance are provided. The value of $\alpha^*$ is affected by three axes: architecture, learning method, and dataset. We do not have thorough experiments on the sensitivity of this value along these axes. However, in Subsection 6.1, we report that there is a slight systematic difference in the estimated slopes between VGG models trained with batch normalization versus without batch normalization. Understanding the sensitivity of this value to architecture, learning method, and dataset is a direction for future work worth investigating, since it could help identify which architecture or algorithm to choose.

We acknowledge that we have not established a fully independent, a priori computation of the box-model dimension $\bar{D}_{\mathcal{M}}$ from architecture alone, even in the modular case, since $\bar{D}_{\mathcal{M}}$ also depends on the learning algorithm. We conjecture that, in the interpolation zone specifically, this dependence on the learning algorithm becomes less significant, so that $\bar{D}_{\mathcal{M}}$ is primarily determined by the architecture; we hope to establish this in future work.

Our theoretical results assume that training and testing points are drawn i.i.d. from the same distribution, with labels generated by the same function $f$. Consequently, our framework does not apply to the classification noise correction setting (Wei et al., 2021), where training labels are generated by $f$ while test labels are generated by a different function $\tilde{f}$. Specifically, the training sets consist of noisy labels $(\mathbf{X}_{train}, f(\mathbf{X}_{train}))$, while the testing sets maintain their original ground-truth labels $(\mathbf{X}_{test}, \tilde{f}(\mathbf{X}_{test}))$. However, if the dataset contains mislabeled data but the same distribution is used for both training and testing, our framework remains applicable.

Finally, while the modular, block-based construction underlying our discrete scale invariance argument reflects common practice in deep learning architectures, we have not empirically verified how closely the resulting log-log linear scaling law, together with its predicted log-periodic oscillations, matches observed

scaling behavior in large-scale trained networks; systematic empirical validation across a range of architectures and datasets would help clarify the practical relevance of the theory. Networks with finer layer structures should also be considered, in contrast to the coarser layer granularity of the published benchmark reports used in Subsection 6.1. Our results are also stated for classification networks; extending the geometric and covering-number arguments developed here to other settings, such as regression, generative modeling, or large language models, is left for future work.

## 6.1 Illustrative examples

Our experiments are designed to explore whether the discrete architectural choices for large networks built from a modular design, such as ResNets, VGGs, Inceptions (GoogLeNets), and ViTs (Vision Transformers), are consistent with the discrete scale invariance property. We leverage this property to attempt to predict the performance of a large network from that of smaller networks, using the solution to the discrete scale invariance equation.

By Lemma 4(ii), the generalization error bound is proportional to the radii of the smallest enclosing balls of the partition regions. Combining this with Part (ii) of Theorem 1, we use the following formulation to estimate $\alpha$ from smaller models with known generalization errors and numbers of parameters:

$$\left(\frac{t_1}{t_2}\right)^{\log_2(1+\alpha)} \approx \left(\frac{g_1}{g_2}\right)^{\log_2(1+\alpha)} \approx \frac{\#W_2}{\#W_1}, \tag{35}$$

where $t_i$, $g_i$ and $\#W_i$ represent the top-1 test error, the generalization error and the number of parameters for model $i$, respectively. This formulation is symmetric in the model indices and can therefore be applied in either direction: predicting a larger model's performance from a smaller one, or vice versa. This approximation assumes the training error is zero or nearly zero. Consequently, the generalization error can be approximated by the expected test error.

Let $\hat{\alpha}$ represent the estimated value of $\alpha$ from the two smallest models. We assess accuracy using two methods, which differ in whether the reference model is updated at each step or held fixed.

**Method A (sequential reference).** This method uses $\hat{\alpha}$ to predict the error of model $i+1$ from the *immediately preceding* model $i$:

$$\left(\frac{t_i}{\hat{t}_{i+1}}\right)^{\log_2(1+\hat{\alpha})} \approx \left(\frac{g_i}{\hat{g}_{i+1}}\right)^{\log_2(1+\hat{\alpha})} \approx \frac{\#W_{i+1}}{\#W_i}. \tag{36}$$

**Method B (fixed reference).** This method uses $\hat{\alpha}$ to predict the error of model $i+1$ from a *fixed reference model* (model 2) throughout:

$$\left(\frac{t_2}{\hat{t}_{i+1}}\right)^{\log_2(1+\hat{\alpha})} \approx \left(\frac{g_2}{\hat{g}_{i+1}}\right)^{\log_2(1+\hat{\alpha})} \approx \frac{\#W_{i+1}}{\#W_2}. \tag{37}$$

To evaluate the accuracy of these predictions, for both methods we compute the relative error (RE):

$$\text{RE} = \frac{|\hat{t}_{i+1} - t_{i+1}|}{t_{i+1}} \times 100\% \approx \frac{|\hat{g}_{i+1} - g_{i+1}|}{g_{i+1}} \times 100\%. \tag{38}$$

We evaluate extrapolation performance in both directions, forward (from smaller to larger models) and backward (from larger to smaller models), to check the consistency of our predictions, using our approach applied to large models built on a modular design with many available model-size data points.

### 6.1.1 ResNet and VGG on ImageNet and CIFAR-100

For the first set, we illustrate these scaling laws using experimental results from ResNet and VGG models on the ImageNet and CIFAR-100 datasets, drawn from various benchmark implementations, related papers, and reports.

In our analysis of VGGs and ResNets on ImageNet, we found that the absolute value of the slope derived from forward prediction for VGGs, $\approx 1.5$, is larger than that for ResNets, $\approx 0.2$. This result suggests that VGGs achieve approximately a 7-fold greater reduction in test error per unit increase in parameters compared to ResNets on a log-log scale. Nevertheless, this efficiency comes at the expense of increased parameter redundancy, as the VGG model has over 100M parameters.

**I. ResNets on ImageNet:**

**Forward:** Table 1 and Figure 2 illustrate the predicted Top-1 Error for the ResNet model on this dataset. As shown in Figure 2, the scaling law indicates that the error bound behaves as a line in the log-log plot of generalization error and the number of parameters. In the log-log plane of error versus the number of parameters, Equation (29) represents a line with slope $-1/\log_2(1 + \alpha)$. The estimated value of $\alpha$ from ResNet-18 and ResNet-34 is 35.14. This results in $\log_2(1 + \hat{\alpha}) = 5.188$ and a slope of $-\frac{1}{\log_2(1+\hat{\alpha})} = -0.19$. This slope suggests, according to Equation (35), that for the test error of a ResNet to be reduced to half its original value, the number of parameters would have to increase to thirty-two times its original size. We use this slope value to derive the predicted Top-1 error presented in the table and figure.

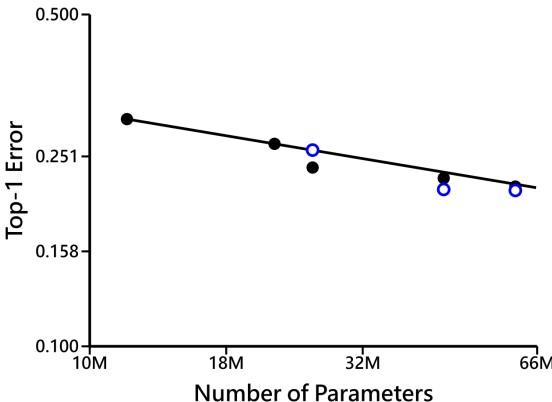

Figure 2: Log-log plot of Table 1. Axes show raw parameter counts and top-1 test errors for ease of interpretation. The line through the two smallest models (ResNet-18, ResNet-34) is used to predict performance of larger models (ResNet-50, ResNet-101, ResNet-152). Solid circles denote actual values; empty circles denote predictions from Method A.

| Model-Depth | Size (M) | Error (%) | A: Pred. Err. (%) | A: RE (%) | B: Pred. Err. | B: RE(%) |
|---|---|---|---|---|---|---|
| ResNet-18 | 11.7 | 30.1 | x | x | x | x |
| ResNet-34 | 21.8 | 26.7 | x | x | x | x |
| ResNet-50 | 25.6 | 23.8 | 25.9 | 8.82 | 25.9 | 8.82 |
| ResNet-101 | 44.5 | 22.6 | 21.4 | 5.31 | 23.3 | 3.1 |
| ResNet-152 | 60.2 | 21.7 | 21.3 | 1.84 | 21.9 | 0.88 |

Table 1: Forward prediction of the top-1 test error for ResNet models on ImageNet, whose validation set contains approximately 6% label errors (Northcutt et al., 2021). Parameter counts and error rates are taken from He et al. (2016) and benchmark re-implementations (Wightman et al., 2021; PyTorch Development Team, 2025); slight variations may arise from differences in training setup. Predictions from smaller models achieve a relative error below 10%, enabling reliable resource assessment for larger models.

**Backward:** The slope is computed from the two largest models, ResNet-101 and ResNet-152.

| Model-Depth | Size (M) | Error (%) | A: Pred. Err. (%) | A: RE (%) | B: Pred. Err. | B: RE (%) |
|---|---|---|---|---|---|---|
| ResNet-18 | 11.7 | 30.1 | 29.03 | 3.5 | 27.05 | 10.13 |
| ResNet-34 | 21.8 | 26.7 | 24.32 | 8.9 | 24.89 | 6.8 |
| ResNet-50 | 25.6 | 23.8 | 24.34 | 2.2 | 24.34 | 2.2 |
| ResNet-101 | 44.5 | 22.6 | x | x | x | x |
| ResNet-152 | 60.2 | 21.7 | x | x | x | x |

Table 2: Backward prediction of the top-1 test error for ResNet models on ImageNet. The absolute value of slope is 1/7.44. Predictions from larger models achieve a relative error below 10%, enabling reliable resource assessment for larger models.

**II. ResNets on CIFAR-100:**

**Forward:** The Top-1 Error, as shown in Table 3, was obtained from the implementation of ResNet models on the CIFAR-100 dataset, as reported by Jafar and Lee (2023). The slope derived from the ResNet-20 and ResNet-34 models is -0.15, with an estimated value of $\alpha$ being 93.2.

| Model-Depth | Size (M) | Error (%) | A: Pred. Err. (%) | A: RE (%) | B: Pred. Err. | B: RE(%) |
|---|---|---|---|---|---|---|
| ResNet-20 | 0.27 | 30.31 | x | x | x | x |
| ResNet-34 | 0.46 | 27.95 | x | x | x | x |
| ResNet-44 | 0.66 | 27.02 | 26.44 | 2.14 | 26.44 | 2.14 |
| ResNet-101 | 1.7 | 23.86 | 23.39 | 1.96 | 22.91 | 3.98 |

Table 3: Forward prediction of the top-1 test error for ResNet models on CIFAR-100, whose validation set contains approximately 5.85% label errors (Northcutt et al., 2021). Parameter counts and error rates are taken from Jafar and Lee (2023); note that parameter counts differ from Table 1 due to variations in layer width. Relative errors for ResNet-44 and ResNet-101 are both below 5%.

**Backward:** The slope is computed from the two largest models, ResNet-101 and ResNet-44.

| Model-Depth | Size (M) | Error (%) | A: Pred. Err. (%) | A: RE (%) | B: Pred. Err. | B: RE |
|---|---|---|---|---|---|---|
| ResNet-20 | 0.27 | 30.31 | 29.98 | x | 30.38 | x |
| ResNet-34 | 0.46 | 27.95 | 28.33 | x | 28.33 | x |
| ResNet-44 | 0.66 | 27.02 | x | x | x | x |
| ResNet-101 | 1.7 | 23.86 | x | x | x | x |

Table 4: Backward prediction of the top-1 test error for ResNet models on CIFAR-100. The absolute value of slope is 1/7.61. Parameter counts and error rates are taken from Jafar and Lee (2023); note that parameter counts differ from Table 1 due to variations in layer width.

**III. VGGs on ImageNet:** In our analysis of the VGG family of models tested on ImageNet, we first examine the least-squares regression lines presented on a log-log scale. The regression lines adhere to scaling laws, displaying slopes of -1.21 for models that utilize Batch Normalization (BN) and -1.11 for those that do not. This difference in slope values indicates that the BN model is slightly more efficient with its parameters than the one without BN, suggesting a faster rate of decrease in test error.

The VGG implementations on ImageNet exhibit a non-negligible training error. However, as long as the training error is comparably small to the test error, we can still apply Eq. (35) to obtain an estimate of the slope for smaller models.

**Forward:** The values of $\alpha$ were found to be 0.551 for models without BN and 0.478 for those with BN. The slope for models without BN is -1.581, while the slope for models utilizing BN is -1.784. The values are slightly lower but qualitatively consistent with those obtained using the least-squares method for three observations. Tables 5 and 6 present the prediction performance results, and the performance differences

shown in the tables primarily result from the implementation of BN. Since there are only three observations, both methods yield identical predictions.

| Model-Depth | Size (M) | Test Error (%) | Pred. Test Error (%) | Relative Error (%) |
|---|---|---|---|---|
| VGG-13 | 133 | 30.1 | x | x |
| VGG-16 | 138 | 28.4 | x | x |
| VGG-19 | 144 | 27.6 | 26.55 | 3.8 |

Table 5: Prediction of the top-1 test error for VGG models (without Batch Normalization) on ImageNet. Parameter counts and error rates are taken from Simonyan and Zisserman (2014) and benchmark re-implementations (Paszke et al., 2019; PyTorch Development Team, 2025); slight variations may arise from differences in training setup.

| Model-Depth | Size (M) | Test Error (%) | Pred. Test Error (%) | Relative Error (%) |
|---|---|---|---|---|
| VGG-13-BN | 133 | 28.4 | x | x |
| VGG-16-BN | 138 | 26.6 | x | x |
| VGG-19-BN | 144 | 25.8 | 24.66 | 4.4 |

Table 6: Forward prediction of the top-1 test error for VGG models with Batch Normalization (Ioffe and Szegedy, 2015) on ImageNet (Paszke et al., 2019; PyTorch Development Team, 2025). Slight variations may arise from differences in training setup.

**Backward:** The slope is computed from the two largest models, VGG-19 and VGG-16.

| Model-Depth | Size (M) | Test Error (%) | Pred. Test Error (%) | Relative Error (%) |
|---|---|---|---|---|
| VGG-13 | 133 | 30.1 | 29.11 | x |
| VGG-16 | 138 | 28.4 | x | x |
| VGG-19 | 144 | 27.6 | x | x |

Table 7: Back prediction of the top-1 test error for VGG models (without Batch Normalization) on ImageNet. The absolute value of slope is $1/1.49$. Parameter counts and error rates are taken from Simonyan and Zisserman (2014) and benchmark re-implementations (Paszke et al., 2019; PyTorch Development Team, 2025).

| Model-Depth | Size (M) | Test Error (%) | Pred. Test Error (%) | Relative Error (%) |
|---|---|---|---|---|
| VGG-13-BN | 133 | 28.4 | 27.31 | 3.29 |
| VGG-16-BN | 138 | 26.6 | x | x |
| VGG-19-BN | 144 | 25.8 | x | x |

Table 8: Backward prediction of the top-1 test error for VGG models with Batch Normalization (Ioffe and Szegedy, 2015) on ImageNet (Paszke et al., 2019; PyTorch Development Team, 2025). The absolute value of slope is $1/1.39$.

In Appendix F, we present more examples. We illustrate these scaling laws using experimental results from InceptionNet and ViT models on the Udacity and ImageNet datasets, respectively, drawn from various benchmark implementations, related papers, and reports.

## 7 Conclusions

In this paper, we have analyzed the scaling laws for deep neural networks and contributed theoretically by linking the geometric properties of a network's decision boundaries to empirical scaling laws. We have introduced the *box-model dimension* to quantify the number of parameters a model needs to describe a shape

at different scales, adapting fractal geometry's box-counting dimension to neural network architecture. Our results establish that the generalization error scales on the order of $1/\#W^{1/\bar{D}_{\mathcal{M}}}$, where $\bar{D}_{\mathcal{M}} = \log_2(1 + \alpha_{\mathcal{M}})$ is the box-model dimension of the model-family, whose value is connected to the learning method and architecture. As a secondary contribution, we have shown that networks constructed with modular design blocks can exhibit a discrete scale-invariance property that suggests a route toward predicting error for larger models from smaller ones, and we have provided preliminary empirical results on ResNet, VGG, Inception, and ViT models to support this hypothesis. Empirically verifying our analysis more broadly is a future direction worth further investigation. We hope this work provides a foundation for future research, including relaxing the semi-algebraic assumption, pursuing larger-scale empirical validation, and extending the analysis to large language models and more general PAC settings.

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

## A   Proof of Lemma 2

*Proof.* Part (i). We show $\partial f \cap \mathcal{X} = \partial f^*$ by proving both inclusions.

Step 1: Consistency of $f$ on $\mathcal{X}$. For any $\mathbf{x} \in \mathcal{X}$, $\pi_{\mathcal{X}}(\mathbf{x}) = \mathbf{x}$, so:

$$f(\mathbf{x}) = f^*(\pi_{\mathcal{X}}(\mathbf{x})) = f^*(\mathbf{x}), \quad \forall \mathbf{x} \in \mathcal{X}. \tag{39}$$

Step 2: $f$ is constant along normal fibers. For any $\mathbf{y} \in U_{\tau/2}(\mathcal{X})$, $\pi_{\mathcal{X}}(\mathbf{y})$ is unique and smooth. By construction, $f(\mathbf{y}) = f^*(\pi_{\mathcal{X}}(\mathbf{y}))$ depends only on the base point $\pi_{\mathcal{X}}(\mathbf{y}) \in \mathcal{X}$, not on the normal displacement $\mathbf{y} - \pi_{\mathcal{X}}(\mathbf{y})$. Therefore $f$ is constant along each normal fiber $\pi_{\mathcal{X}}^{-1}(\mathbf{x})$ for every $\mathbf{x} \in \mathcal{X}$.

Step 3: $\partial f^* \subseteq \partial f \cap \mathcal{X}$. Let $\mathbf{x} \in \partial f^* \subset \mathcal{X}$. Since $\mathcal{X}$ carries the subspace topology from $\mathbb{R}^D$, for any ambient neighborhood $W$ of $\mathbf{x}$, the intersection $V = W \cap \mathcal{X}$ is a nonempty open intrinsic neighborhood of $\mathbf{x}$ in $\mathcal{X}$. By the boundary assumption, $V$ contains points $\mathbf{x}', \mathbf{x}'' \in \mathcal{X}$ with $f^*(\mathbf{x}') \neq f^*(\mathbf{x}'')$. Since $f$ agrees with $f^*$ on $\mathcal{X}$ (Step 1), we have $f(\mathbf{x}') \neq f(\mathbf{x}'')$, with both $\mathbf{x}', \mathbf{x}'' \in W$. Therefore $\mathbf{x} \in \partial f$, and since $\mathbf{x} \in \mathcal{X}$, we conclude $\mathbf{x} \in \partial f \cap \mathcal{X}$.

Step 4: $\partial f \cap \mathcal{X} \subseteq \partial f^*$. Let $\mathbf{x} \in \partial f \cap \mathcal{X}$. Every open intrinsic neighborhood $V$ of $\mathbf{x}$ in $\mathcal{X}$ can be written as $V = W \cap \mathcal{X}$ for some open ambient neighborhood $W$ of $\mathbf{x}$ in $U_{\tau/2}(\mathcal{X})$. Since $\mathbf{x} \in \partial f$, $W$ contains points $\mathbf{y}', \mathbf{y}''$ with $f(\mathbf{y}') \neq f(\mathbf{y}'')$. By continuity of $\pi_{\mathcal{X}}$ and since $\pi_{\mathcal{X}}(\mathbf{x}) = \mathbf{x}$, for $\mathbf{y}', \mathbf{y}''$ sufficiently close to $\mathbf{x}$, we have $\pi_{\mathcal{X}}(\mathbf{y}'), \pi_{\mathcal{X}}(\mathbf{y}'') \in V$, with $f^*(\pi_{\mathcal{X}}(\mathbf{y}')) = f(\mathbf{y}') \neq f(\mathbf{y}'') = f^*(\pi_{\mathcal{X}}(\mathbf{y}''))$. Hence $\mathbf{x} \in \partial f^*$.

Step 5: Preservation of $\gamma_s$ and $\beta_l$. Since $f$ is constant along normal fibers (Step 2) and $\partial f \cap \mathcal{X} = \partial f^*$ (Steps 3–4), the decision regions of $f$ in $U_{\tau/2}(\mathcal{X})$ are precisely the normal tube expansions of the decision regions of $f^*$ in $\mathcal{X}$:

$$\{f(\mathbf{y}) = \mathbf{e}_i\} \cap U_{\tau/2}(\mathcal{X}) = \pi_{\mathcal{X}}^{-1}(\{f^*(\mathbf{x}) = \mathbf{e}_i\}), \quad i = 1, \ldots, l. \tag{40}$$

Therefore $\gamma_s$ and $\beta_l$ defined with respect to $f$ in $U_{\tau/2}(\mathcal{X})$ coincide with those defined with respect to $f^*$ in $\mathcal{X}$.

Part (ii).

Step 1: Reduction to the tube $U_{\delta/2}(\mathcal{X})$. By Assumption (bounded support), $\mathcal{P}$ assigns zero mass outside $U_{\delta/2}(\mathcal{X})$:

$$\int_{\mathbb{R}^D \setminus U_{\delta/2}(\mathcal{X})} \mathcal{P}(\mathbf{x}) \, d\mathbf{x} = 0. \tag{41}$$

Therefore:

$$\int_{\mathbb{R}^D} g(f(\mathbf{y}), \mathcal{M}(\mathbf{y})) \, \mathcal{P}(\mathbf{y}) \, d\mathbf{y} = \int_{U_{\delta/2}(\mathcal{X})} g(f(\mathbf{y}), \mathcal{M}(\mathbf{y})) \, \mathcal{P}(\mathbf{y}) \, d\mathbf{y}, \tag{42}$$

since $\mathcal{P}(\mathbf{y}) = 0$ almost everywhere outside $U_{\delta/2}(\mathcal{X})$. Since $\delta < \tau$, we have $U_{\delta/2}(\mathcal{X}) \subseteq U_{\tau/2}(\mathcal{X})$, so $f$ is defined on the support of $\mathcal{P}$.

Step 2: The right-hand side depends only on intrinsic geometry. For any $\mathbf{y} \in U_{\delta/2}(\mathcal{X}) \subseteq U_{\tau/2}(\mathcal{X})$, $\pi_{\mathcal{X}}(\mathbf{y})$ is unique and smooth, and $f(\mathbf{y}) = f^*(\pi_{\mathcal{X}}(\mathbf{y}))$ depends only on the base point $\pi_{\mathcal{X}}(\mathbf{y}) \in \mathcal{X}$. Therefore the decision boundary structure of $f$ within $U_{\delta/2}(\mathcal{X})$ is governed entirely by the intrinsic geometry of $\partial f^* \subset \mathcal{X}$, established in part (i).

Step 3: Covering arguments scale with $d$. Since the decision boundary structure of $f$ in $U_{\delta/2}(\mathcal{X})$ is intrinsic (Step 2), all covering arguments required to bound the right-hand side scale with the intrinsic covering number of $\mathcal{X}$. By Lemma 1(iii), this satisfies

$$N_{\mathrm{cov}}(\mathcal{X}, \beta_l - \delta/2, \|\cdot\|_2) \leq C \cdot \mathrm{vol}_d(\mathcal{X}) \cdot \beta_l^{-d}, \tag{43}$$

where $C$ depends only on $d$, not on $D$. Therefore all covering arguments scale as $\mathcal{O}(\beta_l^{-d})$, replacing the ambient scaling $\mathcal{O}(\beta_l^{-D})$.

The generalization error of $f$ under $\mathcal{P}$ is entirely determined by its restriction to $U_{\delta/2}(\mathcal{X})$, and all covering arguments scale with the intrinsic dimension $d$ rather than the ambient dimension $D$, under the bounded-reach geometric assumption on $\mathcal{X}$. $\qquad\square$

## B    Proof of Lemma 3

*Proof.* Given that $\beta_l$ is the radius of the largest inscribed ball within all decision regions in the input domain $B_d(\mathbf{0}, 1)$ derived by network $\mathcal{M}$, it follows that for any region, there exists a ball of radius $\beta_l$ contained within that region.

To reach a contradiction, suppose there exists a region containing no training points. Then the center of the inscribed ball of radius $\beta_l$ in that region has no training point within distance $\beta_l$ of it. This contradicts the assumption that the training points form a $\beta_l$-covering of $B_d(\mathbf{0}, 1)$ (Assumption (dense coverage in the manifold)).

$\square$

## C    Proof of Lemma 4

*Proof.* One-hot classifiers are piecewise constant functions, assigning inputs to a single class $i$ within the range of $\{1, \cdots, l\}$. The standard coordinate basis $\mathbf{e}_i$ represents class $i$ in a one-hot coding system. To compute the 0/1 error function, we can use the formula below: for any $f(\mathbf{y}), f(\mathbf{z}) \in \{\mathbf{e}_1, \cdots, \mathbf{e}_l\}$,

$$g(f(\mathbf{y}), f(\mathbf{z})) = \frac{1}{\sqrt{2}} \|f(\mathbf{y}) - f(\mathbf{z})\|_2 = \begin{cases} 1 \text{ if } f(\mathbf{y}) \neq f(\mathbf{z}) \\ 0 \text{ otherwise.} \end{cases} \tag{44}$$

For a given classifier $f$, we define $\eta_f(\mathbf{x})$ as the radius of the largest ball around a point $\mathbf{x}$ in the input domain such that every point inside this ball shares the same class as $\mathbf{x}$; that is,

$$\eta_f(\mathbf{x}) := \sup\{r \geq 0 : B(\mathbf{x}, r) \subseteq R_{f(\mathbf{x})}\}, \tag{45}$$

where $R_{f(\mathbf{x})}$ denotes the decision region of class $f(\mathbf{x})$. Equivalently, $\eta_f(\mathbf{x})$ is the distance from $\mathbf{x}$ to its closest decision boundary. If $\mathbf{x}$ lies on the classification boundary of $f$, this radius is zero, consistent with we have

$$h_f(\mathbf{x}; \|\mathbf{x} - \mathbf{y}\|_2) = g(f(\mathbf{x}), f(\mathbf{y})) = 0 \text{ if } \|\mathbf{x} - \mathbf{y}\|_2 < \eta_f(\mathbf{x}). \tag{46}$$

We can bound $h_f$ with the following radius function:

$$\hat{h}_f(\mathbf{x}; \|\mathbf{x} - \mathbf{y}\|_2) = \begin{cases} 1 \text{ if } \|\mathbf{x} - \mathbf{y}\|_2 \geq \eta_f(\mathbf{x}) \\ 0 \text{ otherwise.} \end{cases} \geq h_f(\mathbf{x}; \|\mathbf{x} - \mathbf{y}\|_2). \tag{47}$$

$\hat{h}_f(\mathbf{x}; \|\mathbf{x} - \mathbf{y}\|_2)$ is a non-decreasing function of $\|\mathbf{x} - \mathbf{y}\|_2$ for any function $f$. In the case where $\mathbf{x}$ is on the classification boundary, $\hat{h}_f(\mathbf{x}; \|\mathbf{x} - \mathbf{y}\|_2)$ will always be equal to 1 for any value of $\mathbf{y} \neq \mathbf{x}$.

The classification network $\mathcal{M}$ is responsible for learning the classifier function $f$ by utilizing $N$ training data pairs $(\mathbf{x}_i, f(\mathbf{x}_i))$. The function of $\mathcal{M}$ is piece-wise constant, dividing the input domain into non-overlapping regions and assigning each region a specific class. The collection of all these regions is referred to as $R_s$. Each region in $R_s$ is associated with two geometric parameters related to balls: the smallest enclosing radius and the largest inscribed radius. The smallest enclosing radius that encompasses all regions in $R_s$ and the largest inscribed radius for all regions in $R_s$ are denoted as $\gamma_s$ and $\beta_l$, respectively.

The training points are $\beta_l$-covering distributed means that every region of $R_s$ must have at least one training point (see Lemma 3). For any input point $\mathbf{x}$ located in region $p$, let $\mathbf{x}_p$ represent a training point within that region. The distance between $\mathbf{x}$ and $\mathbf{x}_p$ will always be less than or equal to $2\gamma_s$, since the radius of the enclosing ball for the region is $\gamma_s$. To calculate the error between $f(\mathbf{x})$ and $\mathcal{M}(\mathbf{x})$ with respect to $\mathbf{x}_p$, we can use the following equation:

$$\begin{aligned} g(f(\mathbf{x}), \mathcal{M}(\mathbf{x})) &= \frac{1}{\sqrt{2}} (\|f(\mathbf{x}) - \mathcal{M}(\mathbf{x}) + f(\mathbf{x}_p) - f(\mathbf{x}_p) + \mathcal{M}(\mathbf{x}_p) - \mathcal{M}(\mathbf{x}_p)\|_2) \\ &\leq \frac{1}{\sqrt{2}} (\|f(\mathbf{x}) - f(\mathbf{x}_p)\|_2 + \|f(\mathbf{x}_p) - \mathcal{M}(\mathbf{x}_p)\|_2 + \|\mathcal{M}(\mathbf{x}_p) - \mathcal{M}(\mathbf{x})\|_2) \\ &\leq g(f(\mathbf{x}), f(\mathbf{x}_p)) + \mathbf{1}_{\text{hist}} \varepsilon_p. \end{aligned} \tag{48}$$

In the second term, we write $\frac{1}{\sqrt{2}}\|f(\mathbf{x}_p) - \mathcal{M}(\mathbf{x}_p)\|_2$ as $\mathbf{1}_{\text{hist}}\varepsilon_p$. When the interpolation condition holds, $f(\mathbf{x}_p) = \mathcal{M}(\mathbf{x}_p)$, corresponding to $\mathbf{1}_{\text{hist}} = 0$; therefore, $\mathbf{1}_{\text{hist}}\varepsilon_p = 0$. In contrast, when the histogram condition holds, $\mathbf{1}_{\text{hist}} = 1$, so $\mathbf{1}_{\text{hist}}\varepsilon_p = \varepsilon_p$. In the last term, since points in the same region $p$ belong to the same class, the term $\|\mathcal{M}(\mathbf{x}_p) - \mathcal{M}(\mathbf{x})\|_2$ is zero.

Denote $\{\mathbf{x}_{p,i}\}_{i=1}^{n_p}$ as the set of $n_p$ training points in region $p$. By applying Eq. (48) to each training point in the region and then computing the average, we can obtain the following result:

$$
\begin{aligned}
g(f(\mathbf{x}), \mathcal{M}(\mathbf{x})) &\leq \frac{1}{n_p} \sum_{i=1}^{n_p} g(f(\mathbf{x}), f(\mathbf{x}_{p,i})) + \mathbf{1}_{\text{hist}}\bar{\varepsilon}_p \\
&\leq \max_i g(f(\mathbf{x}), f(\mathbf{x}_{p,i})) + \mathbf{1}_{\text{hist}}\bar{\varepsilon}_p.
\end{aligned}
\tag{49}
$$

To assess the expected error for probability distribution $\mathcal{P}$, we use Eq. (49) for each point $\mathbf{x}$ in a region $p$, and then sum over all regions of $R_s$ to obtain:

$$
\begin{aligned}
\int_{B_d(\mathbf{0},1)} g(f(\mathbf{x}), \mathcal{M}(\mathbf{x}))\mathcal{P}(\mathbf{x})\,d\mathbf{x} &= \sum_{p \in R_s} \int_p g(f(\mathbf{x}), \mathcal{M}(\mathbf{x}))\mathcal{P}(\mathbf{x})\,d\mathbf{x} \\
&\leq \sum_{p \in R_s} \int_p \max_i g(f(\mathbf{x}), f(\mathbf{x}_{p,i}))\mathcal{P}(\mathbf{x})d\mathbf{x} \\
&\quad + \sum_{p \in R_s} \mathbf{1}_{\text{hist}}\bar{\varepsilon}_p \int_p \mathcal{P}(\mathbf{x})\,d\mathbf{x}.
\end{aligned}
\tag{50}
$$

We now consider the case of $\mathbf{1}_{\text{hist}} = 1$ and $\mathbf{1}_{\text{hist}} = 0$ separately.

The case $\mathbf{1}_{\text{hist}} = 1$ (histogram approximation): in Eq. (50), $\bar{\varepsilon}_p$ represents the mean of the prediction error, which is a constant in region $p$, and the equality holds because $R_s$ partitions the input domain. The left side indicates the generalization error, while the right comprises two integrals. The first integral is the maximum, overall all training points $\mathbf{x}_{p,i}$ in region $p$, of the function $g$. The second term in Eq. (50) can be further derived using the assumption of histogram approximation assumption to relate this term to the average of all training points:

$$
\begin{aligned}
\sum_{p \in P_s} \bar{\varepsilon}_p \int_p \mathcal{P}(\mathbf{x})\,d\mathbf{x} &= \sum_{p \in P_s} \bar{\varepsilon}_p \frac{n_p}{N} = \frac{1}{N} \sum_{p \in P_s} \sum_{i=1}^{n_p} \|f(\mathbf{x}_{p,i}) - \mathcal{M}(\mathbf{x}_{p,i})\|_2 \\
&= \frac{1}{N} \sum_{i=1}^{N} \|f(\mathbf{x}_i) - \mathcal{M}(\mathbf{x}_i)\|_2.
\end{aligned}
\tag{51}
$$

Thus, Eq. (50) can be expressed as:

$$
\begin{aligned}
\int_{B_d(\mathbf{0},1)} g(f(\mathbf{x}), \mathcal{M}(\mathbf{x}))\mathcal{P}(\mathbf{x})\,d\mathbf{x} &\leq \sum_{p \in R_s} \int_p \max_i g(f(\mathbf{x}), f(\mathbf{x}_{p,i}))\mathcal{P}(\mathbf{x})d\mathbf{x} \\
&\quad + \frac{1}{N\sqrt{2}} \sum_i g(f(\mathbf{x}_i), \mathcal{M}(\mathbf{x}_i)).
\end{aligned}
\tag{52}
$$

The case $\mathbf{1}_{\text{hist}} = 0$ (interpolation): In this case, the last term in Eq. (50) vanishes:

$$
\int_{B_d(\mathbf{0},1)} g(f(\mathbf{x}), \mathcal{M}(\mathbf{x}))\mathcal{P}(\mathbf{x})\,d\mathbf{x} \leq \sum_{p \in R_s} \int_p \max_i g(f(\mathbf{x}), f(\mathbf{x}_{p,i}))\mathcal{P}(\mathbf{x})d\mathbf{x}.
\tag{53}
$$

Combining Eqs (52) and (53), we obtain

$$\int_{B_d(\mathbf{0},1)} g(f(\mathbf{x}), \mathcal{M}(\mathbf{x}))\mathcal{P}(\mathbf{x})\, d\mathbf{x} = \sum_{p \in R_s} \int_p g(f(\mathbf{x}), \mathcal{M}(\mathbf{x}))\mathcal{P}(\mathbf{x})\, d\mathbf{x}$$

$$\leq \sum_{p \in R_s} \int_p \max_i g(f(\mathbf{x}), f(\mathbf{x}_{p,i}))\mathcal{P}(\mathbf{x})d\mathbf{x}$$

$$+ \mathbf{1}_{\text{hist}} \frac{1}{N\sqrt{2}} \sum_i g(f(\mathbf{x}_i), \mathcal{M}(\mathbf{x}_i)). \tag{54}$$

We can utilize two facts to calculate an upper bound for the first term in Eq. (54). The first is that $\|\mathbf{x} - \mathbf{x}_{p,i}\|_2 \leq 2\gamma_s$, and the second is that $\hat{h}_f$ is a non-decreasing function, as per Eqs. (47) and (46). By using these facts, we obtain the following inequality:

$$\sum_{p \in R_s} \int_p \max_i g(f(\mathbf{x}), f(\mathbf{x}_{p,i}))\mathcal{P}(\mathbf{x})d\mathbf{x} \leq \sum_{p \in R_s} \int_p \max_i \hat{h}_f(\mathbf{x}; \|\mathbf{x} - \mathbf{x}_{p,i}\|_2)\mathcal{P}(\mathbf{x})d\mathbf{x}$$

$$\leq \int \hat{h}_f(\mathbf{x}; 2\gamma_s)\mathcal{P}(\mathbf{x})d\mathbf{x}$$

$$\leq \int 1([2\gamma_s \geq \eta_f(\mathbf{x})])\mathcal{P}(\mathbf{x})d\mathbf{x}. \tag{55}$$

Here, $1([2\gamma_s \geq \eta_f(\mathbf{x})])$ is an indicator function for $2\gamma_s \geq \eta_f(\mathbf{x})$, and the last inequality is a result of Eq. (47). For function $f$, the value of $\eta_f(\mathbf{x})$ remains fixed. Moreover, the function $1([2\gamma_s \geq \eta_f(\mathbf{x})])$ is an increasing function of $\gamma_s$. Therefore, a network $\mathcal{M}$ with a lower value of $\gamma_s$ can lead to a reduced generalization bound.

(i) The probability mass that contributes to $\int 1([2\gamma_s \geq \eta_f(\mathbf{x})])\mathcal{P}(\mathbf{x})d\mathbf{x}$, which can generate generalization error, can be bounded. The points satisfying $2\gamma_s \geq \eta_f(\mathbf{x})$ are within a distance no greater than $2\gamma_s$ from the classification boundary of $f$, denoted by $\partial f$, as illustrated in the schematic diagram in Figure 1(b). The boundary can be decomposed as the union of semi-algebraic subsets as follows:

$$\partial f = \bigcup_{k=0}^{d-1} f_k = \bigcup_{k=0}^{d-1} \bigcup_i f_k^i,$$

where $f_k^i$ is a $k$-dimensional semi-algebraic set.

**Definition 2** (Tubular Neighborhood). *Let $S \subseteq \mathbb{R}^d$ be a $k$-dimensional embedded submanifold (or, more generally, a stratum of a semi-algebraic set), and let $NS = \{(\mathbf{s}, \mathbf{v}) \in S \times \mathbb{R}^d \mid \mathbf{v} \perp T_\mathbf{s}S\}$ denote its normal bundle. Let*

$$\exp^\perp : NS \to \mathbb{R}^d, \qquad \exp^\perp(\mathbf{s}, \mathbf{v}) = \mathbf{s} + \mathbf{v}$$

*be the normal exponential map, and for $r > 0$ let $NS_r = \{(\mathbf{s}, \mathbf{v}) \in NS \mid \|\mathbf{v}\| \leq r\}$ denote the radius-$r$ disk sub-bundle of $NS$. If $r \leq \tau(S)$, where $\tau(S)$ is the reach of $S$ (Federer, 1959), then $\exp^\perp$ restricted to $NS_r$ is injective. We refer to $\exp^\perp(NS_r)$ as the normal-bundle tube of $S$ at radius $r$.*

*Separately, we define the metric tubular neighborhood of $S$ as*

$$A(r, S) := \{\mathbf{x} \in \mathbb{R}^d \mid \operatorname{dist}(\mathbf{x}, S) \leq r\}, \qquad \operatorname{dist}(\mathbf{x}, S) := \inf_{\mathbf{s} \in S} \|\mathbf{x} - \mathbf{s}\|_2.$$

*When $S$ is a closed manifold without boundary, such as a point (i.e., $S = \overline{S}$), and $r \leq \tau(S)$, the two constructions coincide: $A(r, S) = \exp^\perp(NS_r)$. When $S$ has a nonempty topological frontier $\partial S := \overline{S} \setminus \operatorname{int}(S)$, however, $A(r, S)$ depends only on the closure $\overline{S}$ (since $\operatorname{dist}(\mathbf{x}, S) = \operatorname{dist}(\mathbf{x}, \overline{S})$), while $\exp^\perp(NS_r)$ only sweeps out normal fibers over points of $S$ itself; in this case $A(r, S)$ strictly contains $\exp^\perp(NS_r)$, the difference being the additional volume contributed near $\partial S$. Lemma 5 below quantifies this difference, together with the curvature contribution present for $k \geq 2$.*

Since $\operatorname{dist}(\mathbf{x}, \partial f) = \min_{0 \leq k \leq d-1} \operatorname{dist}(\mathbf{x}, f_k)$, restricting $A(2\gamma_s, \partial f)$ to $B_d(\mathbf{0}, 1)$, the $2\gamma_s$-neighborhood of $\partial f$ is the union of the $2\gamma_s$-neighborhoods of its strata:

$$A(2\gamma_s, \partial f) = \bigcup_{k=0}^{d-1} A(2\gamma_s, f_k), \tag{56}$$

where the $2\gamma_s$-neighborhood of $f_k = \bigcup_i f_k^i$ is defined as

$$A(2\gamma_s, f_k) = \bigcup_i A(2\gamma_s, f_k^i).$$

Since the neighborhoods $A(2\gamma_s, f_k^i)$ of distinct pieces $i$ may overlap, subadditivity of volume gives

$$|A(2\gamma_s, f_k)| \leq \sum_i |A(2\gamma_s, f_k^i)|, \tag{57}$$

with equality only if the pieces' neighborhoods are pairwise disjoint. This step requires no assumption on the reach of $f_k$; it holds for any collection of sets.

To bound each individual term $|A(2\gamma_s, f_k^i)|$, we invoke the following tube volume bound, which does require a reach condition: even when $2\gamma_s$ is small enough that the tube around $f_k^i$ does not self-intersect, $f_k^i$ may still be curved enough that its tube volume exceeds the flat (zero-curvature) estimate, and its frontier may contribute additional volume beyond the flat cylindrical estimate. The reach condition controls the curvature contribution (see Example 1 and Example 2), while the frontier contribution is accounted for separately (see Example 3).

For a piece $f_k^i$ of the stratification, we write $\partial f_k^i := \overline{f_k^i} \setminus f_k^i$ for its topological frontier, and $|\partial f_k^i|$ for its $(k-1)$-dimensional volume. By the frontier condition of a Whitney stratification, $\partial f_k^i$ is a $(k-1)$-dimensional semi-algebraic set, and is in fact a union of pieces already appearing in the stratum $f_{k-1}$.

**Lemma 5** (Tube Volume Bound; Weyl (1939), Federer (1959)). *Let $f_k^i \subseteq B_d(\mathbf{0}, 1)$ be a $k$-dimensional semi-algebraic subset with reach $\tau(f_k^i) > 0$, and let $\partial f_k^i$ denote its topological frontier, with $|\partial f_k^i|$ its $(k-1)$-dimensional volume. Then for any $r \leq \tau(f_k^i)$, the metric tubular neighborhood of $f_k^i$ satisfies*

$$|A(r, f_k^i)| \leq C(r, \tau(f_k^i)) \cdot |f_k^i| \cdot |B_{d-k}(\mathbf{0}, 1)| \cdot r^{d-k} + \tfrac{1}{2} |\partial f_k^i| \cdot |B_{d-k+1}(\mathbf{0}, 1)| \cdot r^{d-k+1}, \tag{58}$$

*where $|f_k^i|$ denotes the $k$-dimensional volume of $f_k^i$, $|B_{d-k}(\mathbf{0}, 1)|$ is the volume of the unit $(d-k)$-ball, and $C(r, \tau(f_k^i)) \geq 1$ is a curvature-correction factor, nonincreasing in $\tau(f_k^i)$, with $C(r, \tau(f_k^i)) \to 1$ as $\tau(f_k^i) \to \infty$ (i.e., as $f_k^i$ becomes flat) and $C(r, \tau(f_k^i)) = 1$ identically whenever $k \leq 1$; we refer to Federer (1959) (see also the tube-volume bounds of Niyogi et al. (2008)) for the explicit form of $C$ for $k \geq 2$. The additive frontier term, with the $\tfrac{1}{2}$ factor, reflects that each frontier point contributes a half-ball cap rather than a full ball, since one hemisphere of directions at $\partial f_k^i$ already lies in the cylindrical part of the tube over $f_k^i$ itself.*

To apply Lemma 5 uniformly across all pieces of the boundary stratification, we introduce a single critical radius $\gamma_0$ and show it is strictly positive.

**Definition 3** (Critical Radius). *Let $f$ be a semi-algebraic classifier with decision boundary $\partial f = \bigcup_{k=0}^{d-1} f_k = \bigcup_{k=0}^{d-1} \bigcup_i f_k^i$, where each $f_k^i$ is a $k$-dimensional semi-algebraic piece of $B_d(\mathbf{0}, 1)$. Define the critical radius*

$$\gamma_0 := \min_{0 \leq k \leq d-1} \min_i \tau(f_k^i), \tag{59}$$

*where $\tau(f_k^i)$ denotes the reach of $f_k^i$.*

**Remark C.1.** *$\gamma_0$ is well-defined and strictly positive: since $f$ is semi-algebraic, its boundary $\partial f$ admits a finite Whitney stratification into finitely many pieces $f_k^i$, each a $k$-dimensional $C^2$-smooth semi-algebraic manifold (possibly with nonempty frontier), for $k = 0, \ldots, d-1$. Each such piece $f_k^i$, being compact and $C^2$, has positive reach $\tau(f_k^i) > 0$ (Federer, 1959). Since there are only finitely many pieces across all strata, the*

*minimum in (59) is attained and $\gamma_0 > 0$. Consequently, whenever $\gamma_s \leq \gamma_0/2$, the condition $2\gamma_s \leq \tau(f_k^i)$ holds for every piece $i$ of every stratum $k$, so Lemma 5 applies uniformly across the entire union bound. Moreover, since $\tau(f_k^i) \geq \gamma_0$ for every piece and $C(r, \cdot)$ is nonincreasing in its second argument, the curvature-correction factor satisfies $C(2\gamma_s, \tau(f_k^i)) \leq C(2\gamma_s, \gamma_0) =: C_0(\gamma_s)$ uniformly across all pieces and strata. We note that the reach condition controls this curvature contribution (see Example 1 and Example 2), while the frontier term $|\partial f_k^i|$ is a separate, purely topological contribution that persists even for flat pieces (see Example 3).*

Combining Eq. (57) with Lemma 5, provided $2\gamma_s \leq \tau(f_k^i)$ for every piece $i$ of every stratum $k$, and writing $|\partial f_k| := \sum_i |\partial f_k^i|$,

$$|A(2\gamma_s, f_k)| \leq \sum_i |A(2\gamma_s, f_k^i)| \leq C_0(\gamma_s) \cdot (2\gamma_s)^{d-k} |B_{d-k}(\mathbf{0}, 1)| \|f_k\| \; + \; \tfrac{1}{2}(2\gamma_s)^{d-k+1} |B_{d-k+1}(\mathbf{0}, 1)| \|\partial f_k\|. \quad (60)$$

By applying the union bound across strata to Eq. (56) — legitimate here since the strata $f_k$ are disjoint by construction of the stratification — and then substituting Eq. (60), we derive a bound for the volume of the $2\gamma_s$-neighborhood of $\partial f$:

$$|A(2\gamma_s, \partial f)| \leq \sum_{k=0}^{d-1} |A(2\gamma_s, f_k)| \leq \sum_{k=0}^{d-1} \left[ C_0(\gamma_s) \cdot (2\gamma_s)^{d-k} |B_{d-k}(\mathbf{0}, 1)| \|f_k\| \; + \; \tfrac{1}{2}(2\gamma_s)^{d-k+1} |B_{d-k+1}(\mathbf{0}, 1)| \|\partial f_k\| \right].$$
$$(61)$$

Consequently,

$$\int 1([2\gamma_s \geq \eta_f(\mathbf{x})]) \mathcal{P}(\mathbf{x}) d\mathbf{x} = c_{\mathcal{P}} |A(2\gamma_s, \partial f)| \leq c_{\mathcal{P}} \sum_{k=0}^{d-1} \left[ C_0(\gamma_s) \cdot (2\gamma_s)^{d-k} |B_{d-k}(\mathbf{0}, 1)| \|f_k\| \right.$$
$$\left. + \; \tfrac{1}{2}(2\gamma_s)^{d-k+1} |B_{d-k+1}(\mathbf{0}, 1)| \|\partial f_k\| \right]. \quad (62)$$

The first equality bounds the probability mass by the volume of the points over the $2\gamma_s$-neighborhood of $\partial f$, where $c_{\mathcal{P}}$ represents the average density of $\mathcal{P}$ distributed across the neighborhood. Since $A(2\gamma_s, \partial f) \subseteq B_d(\mathbf{0}, 1)$, and (recalling our simplification $\delta = 0$, under which $\mathcal{X}$ is identified with $B_d(\mathbf{0}, 1)$, so that $\mathcal{P}$'s bounded-support bounds $K \geq \mathcal{P}(\mathbf{x}) \geq k$ hold for all $\mathbf{x} \in B_d(\mathbf{0}, 1)$), the average density $c_{\mathcal{P}}$ satisfies

$$k = \frac{1}{|A(2\gamma_s, \partial f)|} \int_{A(2\gamma_s, \partial f)} k \, d\mathbf{x} \leq \frac{1}{|A(2\gamma_s, \partial f)|} \int_{A(2\gamma_s, \partial f)} \mathcal{P}(\mathbf{x}) \, d\mathbf{x} = c_{\mathcal{P}} \leq \frac{1}{|A(2\gamma_s, \partial f)|} \int_{A(2\gamma_s, \partial f)} K \, d\mathbf{x} = K,$$
$$(63)$$

so $c_{\mathcal{P}} \in [k, K]$. Since $C_0(\gamma_s) \to 1$ as $\gamma_s \to 0$, the curvature-correction factor does not alter the leading-order approximation in part (ii) below; the frontier correction terms are one order higher in $\gamma_s$ than the leading term of the same stratum $k$, but the same order as the leading term of stratum $k-1$, and likewise do not alter the leading-order approximation, since $|f_{d-1}|$ already dominates as $\gamma_s \to 0$.

(ii) When $\gamma_s$ is small and $|f_{d-1}| \neq 0$, the sum $\sum_{k=0}^{d-1} C_0(\gamma_s) \cdot (2\gamma_s)^{d-k} |B_{d-k}(\mathbf{0}, 1)| \|f_k\|$ in equation (62) can be approximated by the leading term $(2\gamma_s)|B_1(\mathbf{0}, 1)| \|f_{d-1}\| = 4\gamma_s |f_{d-1}|$ with $|B_1(\mathbf{0}, 1)| = 2$ (the length of a segment), since $C_0(\gamma_s) \to 1$ as $\gamma_s \to 0$.

**Example 1** (Intuition via a circle). *Let $S \subseteq \mathbb{R}^2$ be a circle of radius $\rho$, so $|S| = 2\pi\rho$ (its 1-dimensional volume, i.e., arc length), and its reach is $\tau(S) = \rho$: the normal lines at every point of $S$ point toward the center, and they all meet exactly at distance $\rho$, the center itself. Since $S$ is a closed curve without boundary, $\partial S = \emptyset$, so the frontier correction term in Lemma 5 vanishes identically. Since $k = 1$ here, the curvature-correction factor also satisfies $C(r, \rho) = 1$ identically, per Lemma 5.*

*For a tube radius $r < \rho = \tau(S)$, the tube $A(r, S)$ is the annulus between radius $\rho - r$ and radius $\rho + r$, with exact area*

$$|A(r, S)| = \pi(\rho + r)^2 - \pi(\rho - r)^2 = 4\pi\rho r = |S| \cdot 2r,$$

which matches the flat estimate $|S| \cdot |B_1(\mathbf{0},1)| \cdot r = (2\pi\rho)(2)(r)$ exactly, consistent with $C(r,\rho) = 1$. This exactness illustrates the mechanism: the annulus's outer edge (circumference $2\pi(\rho + r)$) is longer than $S$, and its inner edge (circumference $2\pi(\rho - r)$) is shorter than $S$; these two effects exactly cancel for a curve ($k = 1$), leaving the flat estimate untouched.

Now suppose instead $r \geq \rho = \tau(S)$. The inner boundary of the would-be annulus, at radius $\rho - r \leq 0$, has collapsed past the center: the normal fibers on opposite sides of the circle now overlap. The tube is no longer an annulus but a full disk of radius $\rho + r$, with area $\pi(\rho + r)^2 < |S| \cdot 2r$ for $r$ sufficiently larger than $\rho$. Here the flat estimate actually overcounts, since the overlapping fibers are counted only once in the true tube but would be counted twice in the flat product estimate — illustrating why $r \leq \tau(S)$ is also needed to avoid double-counting from self-intersection.

**Example 2** (A sphere reveals the curvature correction). Let $S \subseteq \mathbb{R}^3$ be a sphere of radius $\rho$, so $k = 2$, $d = 3$, $|S| = 4\pi\rho^2$ (its 2-dimensional volume, i.e., surface area), and its reach is $\tau(S) = \rho$. Since $S$ is a closed surface without boundary, $\partial S = \emptyset$, so the frontier correction term vanishes identically, as for the circle. Unlike the circle, however, $k = 2$ here, so Lemma 5 does not guarantee $C(r,\rho) = 1$.

For a tube radius $r < \rho = \tau(S)$, the tube $A(r,S)$ is the spherical shell between radius $\rho - r$ and radius $\rho + r$, with exact volume

$$|A(r,S)| = \tfrac{4}{3}\pi(\rho + r)^3 - \tfrac{4}{3}\pi(\rho - r)^3 = 8\pi\rho^2 r + \tfrac{8}{3}\pi r^3 = |S| \cdot 2r + \tfrac{8}{3}\pi r^3,$$

which exceeds the flat estimate $|S| \cdot |B_1(\mathbf{0},1)| \cdot r = 8\pi\rho^2 r$ by the strictly positive term $\tfrac{8}{3}\pi r^3$. This confirms that, unlike the $k = 1$ case, the flat estimate alone is not a valid upper bound once $k \geq 2$: the curvature-correction factor $C(r,\rho) > 1$ is genuinely needed here, and Lemma 5 accounts for this via $C(r,\rho) \cdot 8\pi\rho^2 r$, with $C(r,\rho) \geq 1 + \frac{r}{3\rho}$ required to dominate the exact volume above.

**Example 3** (A segment in the plane). Let $S = (a,b) \times \{0\} \subset \mathbb{R}^2$ be an open line segment of length $L = b - a$, so $k = 1$, $d = 2$, and $\partial S$ consists of the two endpoints, $|\partial S| = 2$. Since $S$ is flat, $\tau(S) = \infty$, so Lemma 5 applies for every $r > 0$, with $C(r,\infty) = 1$. It gives

$$|A(r,S)| \leq |S| \cdot |B_1(\mathbf{0},1)| \cdot r + \tfrac{1}{2} \cdot 2 \cdot |B_2(\mathbf{0},1)| \cdot r^2 = 2rL + \pi r^2,$$

using $|B_1(\mathbf{0},1)| = 2$ and $|B_2(\mathbf{0},1)| = \pi$. This matches the exact area of the metric tube (a rectangle capped by two half-disks) with equality, confirming the bound is tight in this flat, boundary-only case.

$\square$

# D    Box-model dimensions of shallow and deep networks

We develop DNNs by increasing depth, in contrast to the wider configurations discussed by Sharma and Kaplan (2022).

Assume that the training points are distributed within a unit cube $[0,1]^d$. The unit cube can be structured as a grid composed of hypercubes, each with a side length of $s < 1$. The number of hypercubes, denoted as $N_s$, is given by $(s)^{-d}$. Sharma and Kaplan (2022) suggest that the number of parameters $\#S_s$ of a network $\mathcal{S}_s$ required to realize this grid is proportional to $N_s$.

Analogously, if the side length of hypercubes in the grid becomes $2s < 1$, the number of hypercubes is $N_{2s} = (2s)^{-d}$, and the number of parameters $\#S_{2s}$ of a network to realize this grid is proportional to $N_{2s}$.

Following this reasoning, when reducing the side length from $2s$ to $s$, the number of parameters required for the network $\mathcal{S}_s$ is given by $\#S_s = c(s)^{-d} = 2^d \#S_{2s}$, where $c$ is the proportionality constant. Thus, halving the side length increases the required number of parameters by a factor of $2^d$.

This exponential increase in parameter counts results from widening the network while keeping the depth fixed. However, we argue that the parameters used to realize the grid with side length $2s$ can be reused to construct the finer grid with side length $s$, which the approach of Sharma and Kaplan (2022) does not consider.

Alternatively, the grid configuration with side length $s$ can be constructed from two grids with side length $2s$, where the second is obtained by translating the first by $s$ units along each axis. If a network realizing a grid of side length $2s$ can be reused and augmented with this translation, the number of parameters required for the finer grid with side length $s$ can be substantially reduced compared to training a new network from scratch.

Suppose we have a sub-network $\mathcal{N}_{2s}^1$ that consists of $\frac{1}{2s}$ parallel and equally spaced hyperplanes with respect to the first component of the coordinate system of $d$ components. From those hyperplanes, we can generate $\frac{1}{s}$ parallel and equally spaced hyperplanes by adding a layer to $\mathcal{N}_{2s}^1$ as follows:

$$\mathcal{N}_s^1 = \mathrm{ReLU}\left[\begin{array}{c}\mathcal{N}_{2s}^1 \\ \mathcal{N}_{2s}^1 + \begin{bmatrix}-s\\0\\ \vdots \\0\end{bmatrix}\end{array}\right] = \mathrm{ReLU}\ \mathbf{A}_s\mathcal{N}_{2s}^1. \tag{64}$$

The upper component in $\mathbf{A}_s$ is the identity matrix $\mathbf{I}$, which generates the same hyperplanes as those in $\mathcal{N}_{2s}^1$. The bottom mapping involves translating by $-s$ along the first coordinate axis, resulting in another set of hyperplanes, each of which is shifted by $s$ units toward the origin relative to the corresponding hyperplane in $\mathcal{N}_{2s}^1$. The ReLU layer clips the output to $[0,1]^d$.

By concatenating the outputs of $\mathcal{N}_{2s}^i$ for $i = 1, \cdots, d$, we obtain the network that generates the grid configuration of side length $s$:

$$\mathcal{N}_s = \begin{bmatrix}\mathcal{N}_{2s}^1 \\ \vdots \\ \mathcal{N}_{2s}^d\end{bmatrix} = \mathrm{ReLU}\left(\mathrm{diag}(\mathbf{A}_s, \cdots, \mathbf{A}_s)\right)\mathcal{N}_{2s}, \tag{65}$$

where $\mathrm{diag}(\mathbf{A}_s, \cdots, \mathbf{A}_s)$ is the block diagonal matrix with each block $\mathbf{A}_s$. $\mathcal{N}_s$ produces a grid partition of side length $s$ over the input domain $[0,1]^d$. Figure 3 shows how the network $\mathcal{N}_s$ is constructed within the domain $[0,1]^2$, as described in Eqs. (64) and (65).

This construction shows that adding a layer increases the number of parameters linearly with respect to $d$. This is captured by $\#W_s = \#W_{2s} + dc \leq (1 + \alpha)\#W_{2s}$, where $c$ denotes the number of parameters for $\mathbf{A}_s$, $\alpha > 0$, and $\#W_s$ and $\#W_{2s}$ denote the number of parameters of $\mathcal{N}_s$ and $\mathcal{N}_{2s}$, respectively. This linear increase in parameter counts stands in sharp contrast to the exponential growth observed in the network of Sharma and Kaplan (2022).

Next, we compare the mean absolute test errors. Assume that each partition region of $\mathcal{N}_s$ in $[0,1]^d$ contains at least one training point. Let $\mathbf{x}$ and $\mathbf{x}_i$ be two points within a hypercube of side length $s$, with $\|\mathbf{x}-\mathbf{x}_i\| \leq \sqrt{d}$. By the geometry of the hypercube, $\sqrt{d}s$ represents the maximum distance between two points inside a hypercube of side length $s$.

Define $f : \mathbb{R}^d \to \mathbb{R}$ as the target function with Lipschitz constant $K_f$, where $|f(\mathbf{x}) - f(\mathbf{y})| \leq K_f\|\mathbf{x} - \mathbf{y}\|_2$. We suppose $\mathcal{N}_s$ interpolates $f$ at the training points; i.e., $\mathcal{N}_s(\mathbf{x}_i) = f(\mathbf{x}_i)$. Since $\mathcal{N}_s$ is a piecewise constant function over a partition of hypercubes and each hypercube contains at least one training point, if $\mathbf{x}$ and $\mathbf{x}_i$ are in the same hypercube, then $\mathcal{N}_s(\mathbf{x}) = \mathcal{N}_s(\mathbf{x}_i)$. This leads us to the bound:

$$|\mathcal{N}_s(\mathbf{x}) - f(\mathbf{x})| = |\mathcal{N}_s(\mathbf{x}_i) - f(\mathbf{x})| = |f(\mathbf{x}_i) - f(\mathbf{x})| \leq K_f\|\mathbf{x}_i - \mathbf{x}\|_2 \leq K_f\sqrt{d}s.$$

Consequently, if we let $2^l s = 1$, where $l$ denotes the number of refinement levels, the test error can be bounded by:

$$\int_{[0,1]^d} |f(\mathbf{x}) - \mathcal{N}_s(\mathbf{x})|d\mathbf{x} \leq K_f\sqrt{d}s = K_f\sqrt{d}(1/2^l). \tag{66}$$

By unfolding the recurrence equation $\#W_s \leq (1 + \alpha)\#W_{2s}$ $l$ times and using the boundary condition $\#W_1 = 1$, corresponding to a single hypercube over $[0,1]^d$, we can determine the number of parameters for

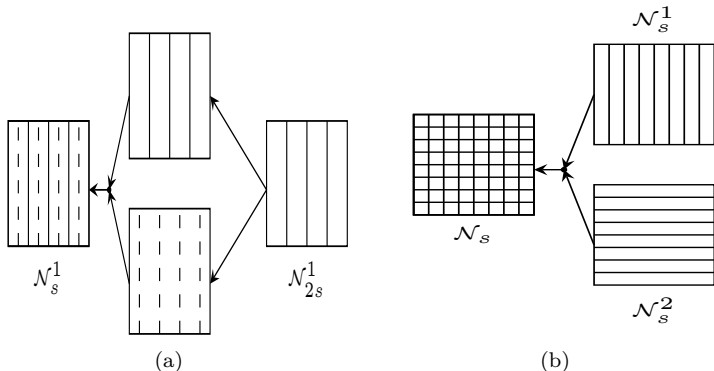

Figure 3: Schematic diagrams in (a) and (b) illustrate how the network $\mathcal{N}_s$ is formed by adding an additional layer to $\mathcal{N}_{2s}$, with its domain in $[0,1]^2$. Subfigure (a) depicts the configurations along the first coordinate, transitioning from $\mathcal{N}_{2s}^1$ to $\mathcal{N}_s^1$. Here, the distance between the parallel hyperplanes in $\mathcal{N}_{2s}^1$ is $2s$, while the distance for the hyperplanes in $\mathcal{N}_s^1$ is $s$. The middle top subfigure corresponds to $\mathcal{N}_{2s}^1$, whereas the middle bottom subfigure shows the configuration of $\mathcal{N}_{2s}^1 + [-s\ 0]^\top$. The configuration of $\mathcal{N}_s^1$ is obtained by concatenating those in $\mathcal{N}_{2s}^1$ and $\mathcal{N}_{2s}^1 + [-s\ 0]^\top$, as defined by Eq. (64). Subfigure (b) illustrates how the configurations of $\mathcal{N}_s^1$ and $\mathcal{N}_s^2$ are concatenated to obtain the configuration of $\mathcal{N}_s$, as defined by Eq. (65). The domain partition of $\mathcal{N}_s$ corresponds to the Cartesian product of the domain partitions of $\mathcal{N}_s^1$ and $\mathcal{N}_s^2$, as stated by Hwang and Tung (2023).

network $\mathcal{N}_s$ as follows:

$$\#W_s \le (1+\alpha)^l \#W_1 \le (1+\alpha)^l. \tag{67}$$

If we substitute $2^l = (1+\alpha)^{l/\log_2(1+\alpha)}$ into Eq. (67), we obtain: $2^l \ge \#W_s^{1/\log_2(1+\alpha)}$. Then, if we substitute this expression for $2^l$ into Eq. (66), the test error for the network $\mathcal{N}_s$ is bounded by:

$$\frac{K_f\sqrt{d}}{\#W_s^{1/\log_2(1+\alpha)}}. \tag{68}$$

The absolute value of the slope in the log-log plot is $1/\log_2(1+\alpha)$.

In contrast, using the recurrence for the network of Sharma and Kaplan (2022): $\#S_s = 2^d \#S_{2s}$, with boundary condition $\#S_1 = 1$, we obtain $\#S_s = 2^{ld}\#S_1 = 2^{ld}$. Substituting $2^l = (\#S_s)^{1/d}$ into Eq. (66), the test error for the network $\mathcal{S}_s$ is bounded by

$$\frac{K_f\sqrt{d}}{\#S_s^{1/d}}. \tag{69}$$

The slope's absolute value is $1/d$, as noted in Sharma and Kaplan (2022).

**Remark D.1.** *A bound that requires neither interpolation at training points nor a $\gamma_s$-covering of the input domain, in contrast to Eq. (66), is given in Eq. (2.2) of Sharma and Kaplan (2022), where $L = \int_{[0,1]^d}(f(\mathbf{x}) - \mathcal{N}_s(\mathbf{x}))^2 d\mathbf{x} \lesssim K_f^2 s^2 d$. Intuitively, the error comes entirely from how much $f$ can vary within a single hypercube. The Lipschitz condition bounds that variation by the diameter of the hypercube, $\sqrt{d}s$. The integral then averages this worst-case error over the whole domain, and since the domain has unit volume, the average equals the pointwise worst-case bound.*

## E  Proof of Proposition 1

*Proof.* Because the two networks $[I+\rho M_L]\circ[I+\rho M_{L-1}]\circ\cdots\circ[I+\rho M_1]$ and $[\rho M_L]\circ[\rho M_{L-1}]\circ\cdots\circ[\rho M_1]$ share the same input domain partition Hwang and Tung (2023), we can simplify our notation for convenience. If

our focus is on the input domain partition, we define:

$$\mathcal{M}_L^0 = [\rho M_L] \circ [\rho M_{L-1}] \circ \cdots \circ [\rho M_1].$$

We will construct $\mathcal{M}_l^0 = [\rho M_l] \cdots [\rho M_1]$ layer by layer, ensuring that it satisfies the condition in Eq. (15) with a constant value of $\alpha$ maintained throughout the network's depth.

The function of $\mathcal{M}_L^0$ is piece-wise continuous. Pascanu et al. (2013) previously discussed the machine's capability for function approximations, focusing on the number of regions generated for function approximation. If $P_l^0$ is the finest input domain partition of $\mathcal{M}_l^0$, where each partitioning region of $P_l^0$ contains no subregions, then $|P_l^0|$ represents the size of $P_l^0$. When considering $P_l^0$, each partitioning region is a polytope, an affine linear mapping domain. Additionally, we note that $P_{l+1}^0$ is a refined partition of $P_l^0$, meaning that every region in $P_{l+1}^0$ is within a region in $P_l^0$. This refinement of partitioning regions in $P_l^0$ sets up a tree-like partitioning of $P_l^0$, where each region in $P_{l+1}^0$ has a unique parent region in $P_l^0$.

Within $P_L^0$, every partitioning region forms a polytope, essentially the intersection of half-planes. The "diameter" of a region denotes the longest distance between two points within the region. The maximum diameter among all polytopes in a partition is equivalent to the diameter of the smallest enclosing ball of the partition. Calculating the diameter of a polytope according to Frieze and Teng (1994) can be intricate for polytopes of arbitrary dimension and facet number. Here, we are developing a tractable algorithm to partition a polytope and reduce the diameters of the resulting sub-polytopes to a constant fraction of the original value.

The algorithm begins by dividing a polytope $p$ in $P_L^0$ via a cutting hyperplane passing through an anchor point in $p$ into two sub-polytopes. This method is then applied recursively to each sub-polytope. As a result, polytope $p$ is the root of a tree-like structure, with intermediate and leaf nodes representing the sub-polytopes. The sub-polytopes at the leaf nodes that refine $p$ will have a diameter smaller than half that of the root polytope. Consequently, the collection of sub-polytopes at the leaf nodes of all trees forms the partition $P_{L+1}^0$, with the diameter to be less than half that of $P_L^0$.

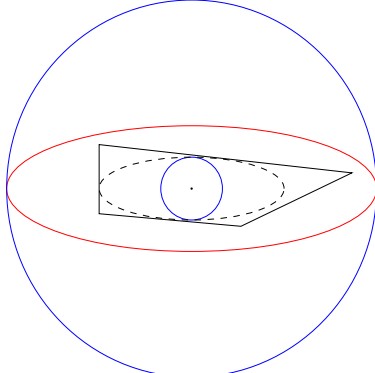

Figure 4: The dashed ellipsoid represents the maximum inscribed ellipsoid of polygon $p$. As per Eq. (71), the primary axis of the ellipsoid has a radius of $\eta$, while the minor axis has a radius of $\tilde{\eta}$, and their ratio is given by $\tilde{\eta}/\eta \geq \zeta > 0$. The center of the ellipsoid corresponds to the highlighted anchor point $\mathbf{a}^p$ of the polygon. The outer ellipsoid, shown in red and expanded by a factor of two at the boundary of the maximum inscribed ellipsoid, encloses the polygon. Enclosed within the polygon is a blue inner circle with a radius of $\tilde{\eta}$, and the polygon itself is enclosed by a blue outer circle with a radius of $2\eta$; hence, $B_2(\mathbf{a}^p, \tilde{\eta}) \subseteq p \subseteq B_2(\mathbf{a}^p, 2\eta)$.

The anchor point is the center of the maximum inscribed ellipsoid within the polytope. By Boyd and Vandenberghe (2004), we can determine this point using convex optimization methods. Additionally, we can utilize properties that approximate the volume of a polytope from both inside and outside at the anchor point, as described in Grötschel et al. (2012) to bound the radius of the smallest enclosing ball of a polytope in $P_{L+1}^0$. Let $\mathbf{a}^p \in \mathbb{R}^d$ be the anchor point of the polytope $p$. Any hyperplane passing through $\mathbf{a}^p$ divides the polytope into two sub-polytopes, with $q$ being one of them. The volume-reduction property shows a

reduction in volume between $p$ and $q$, as demonstrated by the inequality

$$\text{vol}(q) \leq \left(1 - \frac{1}{d}\right) \text{vol}(p). \tag{70}$$

Additionally, if polytope $p$ is convex (or its convex hull if it's not), then the volume of $p$ can be effectively approximated from the outside by an ellipsoid enlarging a constant factor of the inner maximum inscribed ball of $p$; specifically, the outer ellipsoid of $p$ represents the boundary of the inner ellipsoid, expanded by a factor $d$ around the anchor point.

Let $\eta > 0$ represent the radius of the principal axis of the inner maximum volume inscribed ellipsoid of $p$, and $\tilde{\eta} > 0$ denote the smallest radius of this ellipsoid. Assume the relationship $\tilde{\eta}/\eta \geq \zeta > 0$ holds, where $\zeta$ is a predetermined value. Then, we can express the approximation of polytope volume from inside and outside by balls in the following inequality:

$$\zeta^d \eta^d \, \text{vol} \, B_d(\mathbf{0}, 1) \leq \tilde{\eta}^d \, \text{vol} \, B_d(\mathbf{0}, 1) \leq \text{vol}(p) \leq (d\eta)^d \, \text{vol} \, B_d(\mathbf{0}, 1). \tag{71}$$

The first inequality follows the condition $\tilde{\eta}/\eta \geq \zeta$. The second inequality approximates the volume of $p$ from the inside using a ball of radius $\tilde{\eta}$, applying the formula $\text{vol} B_d(\mathbf{0}, r) = r^d \text{vol} B_d(\mathbf{0}, 1)$. The final inequality provides an approximation from the outside, where $d\eta$ represents the radius of an enclosing ball around $p$. Figure 4 offers a schematic illustration of Eq. (71).

Let's consider sub-polytopes formed by cutting $p \in P_L^0$ with a hyperplane passing the anchor point of $p$. These sub-polytopes are children of the root polytope $p$ in a binary branch. This process can be repeated for each sub-polytope, creating a binary tree of depth $k$, with at most $2^k$ leaves, each leaf sub-polytope connected to a path rooted at $p$. We can label the sub-polytope in a path at depth one as $q_1$, the sub-polytope in the path at depth two as $q_2$, and so on, as shown in Figure 5, which provides a schematic description of a path of sub-polytopes in the tree and their hyperplane cuttings. To ensure a child's sub-polytope has a non-empty volume, we assume that all eigenvalues of the largest inscribed ellipsoids of any sub-polytope $q_i$ are greater than zero, and the ratio of the smallest radius to the largest radius of a maximum inscribed ellipsoid is not less than $\zeta$. Let's denote $\eta_k$ and $\tilde{\eta}_k$ as the largest and smallest radii of the principal axes of the maximum volume inscribed ellipsoid of $q_k$. Using Eqs. (70) and (71), we obtain the inequality

$$\zeta^d \eta_k^d \text{vol} \, B_d(\mathbf{0}, 1) \leq \text{vol}(q_k) \leq (1 - \frac{1}{d})^k \text{vol} \, (p). \tag{72}$$

As we increase the depth $k$, the volume of $q_k$ decreases to zero. This implies that the sequence $\{\eta_k\}$ decreases to zero as $k$ increases, satisfying the inequality

$$\eta_k \leq \left[\frac{(1 - \frac{1}{d})^k \text{vol}(p)}{\zeta^d \text{vol} \, B_d(\mathbf{0}, 1)}\right]^{\frac{1}{d}}. \tag{73}$$

Thus, for each path from root polytope $p$ to leaf sub-polytope $q_k$ of the binary tree, we can find a sufficiently large $k$ such that $(d\eta_k) \leq \frac{\gamma_p}{2}$, where $\gamma_p$ is the radius of the smallest enclosing ball of $p$, and $q_k$ is enclosed by a ball of radius $d\eta_k$. The number of hyperplane cuts required to create the path from $p$ to $q_k$ is equal to $k$, the depth of the path, with each hyperplane cut requiring $d + 1$ parameters.

For a binary tree of depth $k - 1$, the number of nodes in the tree is fewer than $2k$. Based on the construction, each node contains one hyperplane with $d + 1$ parameters that divides the polytope associated with the node into two sub-polytopes. Although the number of paths grows exponentially as $2^k$, the number of hyperplane cuts — and hence the parameter count — is determined solely by the number of nodes in the tree, which grows only exponentially as $2k$. Thus, despite the exponential number of paths, few than $2k(d+1)$ parameters are needed to reduce the enclosing radius of $p$ to half its value for all paths in the tree.

Note that each region in $P_l^0$ has a specific depth for its own binary tree. We define $\omega \in \mathbb{N} \cup \{0\}$ as the maximum depth of the binary tree corresponding to any polytope in $P_l^0$ for $l \leq L$ (thus, $\omega \geq k$). Hence, the parameter

bound $2k(d+1)$ from the previous argument extends to $2\omega(d+1)$, where $\omega$ serves as the uniform depth bound across all regions in $P_l^0$. Using $\omega$, the number of polytopes in $P_{l+1}^0$ is at most $|P_{l+1}^0| \leq 2^\omega |P_l^0| \leq 2^{l\omega}$ with $|P_0^0| = 1$. Since the subdivision of a polytope requires at most $2\omega(d+1)$ parameters and there are at most $2^\omega$ polytopes, the number of parameters needed to generate $P_{l+1}^0$ from $P_l^0$ is bounded by $2\omega(d+1)2^{l\omega}$.

The number of parameters in $\mathcal{M}_{l+1}^0$, denoted by $\#W_{l+1}$, is equal to the number of parameters in $\mathcal{M}_l^0$, denoted by $\#W_l$, plus the number of parameters required to obtain $P_{l+1}^0$ from $P_l^0$, which is bounded by $2\omega(d+1)2^{l\omega}$. With the base case $\#W_0 \leq 2\omega(d+1)$, we obtain

$$\#W_{l+1} \leq \#W_l + 2\omega(d+1)2^{l\omega} \leq (1+\alpha)\#W_l. \tag{74}$$

Recursively expanding $\#W_l$ in the above inequality for $\omega \in \mathbb{N}$, we obtain

$$\#W_l \leq 2\omega(d+1)\sum_{i=0}^{l-1} 2^{i\omega} = 2\omega(d+1)\frac{2^{l\omega}-1}{2^\omega-1}. \tag{75}$$

Since $1+\alpha \geq \frac{\#W_{l+1}}{\#W_l}$, taking the ratio of consecutive terms gives

$$1+\alpha \geq \frac{2^{(l+1)\omega}-1}{2^{l\omega}-1}. \tag{76}$$

For large $l$, $\frac{2^{(l+1)\omega}-1}{2^{l\omega}-1} \approx 2^\omega$, and hence $\alpha \gtrsim 2^\omega - 1$. Since $\omega \geq 1$, $\alpha \geq 1$ at large $l$.

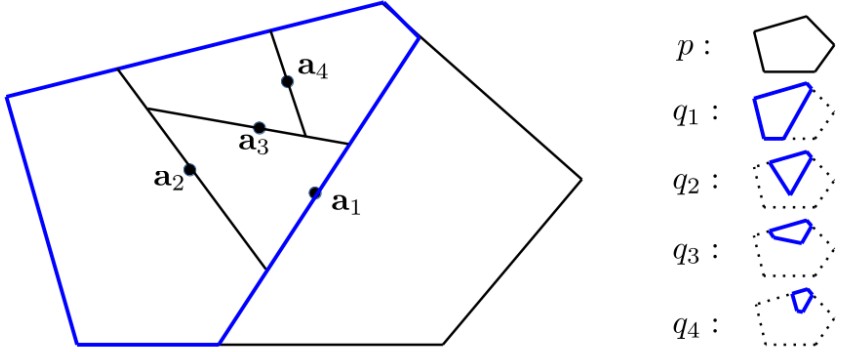

Figure 5: The convex polygon $p$ can be partitioned into sub-polytopes using sequences of hyperplane cuttings, leading to a binary tree structure with $p$ at the root. The root $p$ is divided into two sub-polygons by a hyperplane passing through the anchor point $\mathbf{a}_1$, the center of the maximal inscribed ball of $p$. The sub-polygon $q_1$, colored in blue, is further divided using anchor points $\mathbf{a}_2$, $\mathbf{a}_3$, and $\mathbf{a}_4$, corresponding to the centers of the maximal inscribed balls for sub-polygons $q_2$, $q_3$, and $q_4$, respectively. It is worth noting that the volume of any sub-polygon retains a constant fraction of its parent polygon, as defined by Eq. (70).

To complete the construction, we need to translate all cutting hyperplanes of the trees into a new layer. By induction, suppose we obtain $\mathcal{M}_l^0$. We denote the restriction of $\mathcal{M}_l^0$ on the polytope $p \in \mathcal{P}_l^0$ as $\mathcal{M}_l^0|p$. Let $q \in \mathcal{P}_{l+1}^0$ represent a sub-polytope of $p$. We denote $\mathcal{M}_{l+1}^0|q = \rho\mathbf{A}_{q,p}\mathcal{M}_l^0|p$ the equations define polytope $q$, where $\rho\mathbf{A}_{q,p}$ is the affine mapping derived from the sequence of cutting hyperplanes used to construct $q$ from $p$. In the derivation, we can replace $\rho$ by a diagonal matrix containing entries 0 or 1. The values in this matrix are determined by the values of $\mathcal{M}_l^0|p$. The overall affine linear mapping $M_{l+1}$ is then constructed by stacking the arrays of $\mathbf{A}_{q,p}$ generated from each polytope $p$ in $\mathcal{P}_l^0$.

$\square$

# F   Illustrative examples: InceptionNets on Udacity and ViTs on ImageNet

**IV. InceptionNets on Udacity:**

Nadella et al. (2024) measures the performance of Inception architectures on the Kaggle SAP dataset of approximately 97,330 road images with corresponding steering angles, predicting continuous-valued steering angles from road images. While our analysis is primarily designed for classification, we apply our method to this regression task to demonstrate its generalizability. The MSE, computed over all test images, serves as the prediction error reported in the following table.

**Forward:**

| Model | Size (M) | MSE | A: Pred. MSE | A: RE (%) | B: Pred. MSE | B: RE(%) |
|---|---|---|---|---|---|---|
| InceptionNet | 13 | 0.06044 | x | x | x | x |
| InceptionNet a | 14.5 | 0.05945 | x | x | x | x |
| InceptionNet b | 15.5 | 0.05849 | 0.05885 | 0.62 | 0.05885 | 0.62 |
| InceptionNet c | 17 | 0.05846 | 0.057649 | 1.39 | 0.05805 | 0.7 |
| InceptionNet d | 17.6 | 0.05961 | 0.05815 | 2.5 | 0.05775 | 3.12 |
| InceptionNet e | 20.7 | 0.05802 | 0.05817 | 0.26 | 0.05636 | 2.86 |
| InceptionNet f | 21.7 | 0.05907 | 0.05761 | 2.47 | 0.05602 | 5.16 |

Table 9: Forward predicted MSE for the InceptionNet family of models on the Udacity dataset. The box-model dimension $\log_2(1 + \alpha)$, estimated from the first two models, is 6.612.

**Backward:** InceptionNet-f is not used as a base for prediction because its MSE is larger than that of InceptionNet-e. The slope is calculated from the models InceptionNet-e and InceptionNet-d.

| Model | Size (M) | MSE | A: Pred. MSE | A: RE (%) | B: Pred. MSE | B: RE(%) |
|---|---|---|---|---|---|---|
| InceptionNet | 13 | 0.06044 | 0.060541 | 0.16 | 0.06270 | 3.7 |
| InceptionNet a | 14.5 | 0.05945 | 0.0591 | 0.5 | 0.06157 | 3.5 |
| InceptionNet b | 15.5 | 0.05849 | 0.0594 | 1.56 | 0.06088 | 0.4 |
| InceptionNet c | 17 | 0.05846 | 0.06 | 0.65 | 0.06 | 0.65 |
| InceptionNet d | 17.6 | 0.05961 | x | x | x | x |
| InceptionNet e | 20.7 | 0.05802 | x | x | x | x |

Table 10: Backward Predicted MSE for the InceptionNet family of models on the Udacity dataset. The box-model dimension $\log_2(1 + \alpha)$, estimated from the last two models, is 6.

## V. ViT on ImageNet with JFT pretraining:

Similarly, Dosovitskiy et al. (2020) report the performance of Vision Transformer (ViT) architectures pre-trained on large-scale datasets and evaluated on downstream benchmarks such as ImageNet, CIFAR-100, and VTAB. Like ResNets and Inception networks, ViT is built from stacked, self-contained transformer modules, making it consistent with the modular design underlying our discrete scale invariance argument. We apply our method to the reported top-1 accuracy of ViT models to further test our approach on a transformer-based architecture, distinct from the convolutional networks considered elsewhere in this section. The top-1 error, computed on the corresponding evaluation benchmark, serves as the prediction error reported in the following table.

**Forward:** The slope is calculated from the two smallest models: VIT-B/16 and VIT-L/16.

| Model(JFT-) | Size (M) | Error (%) | A: Pred. Err. (%) | A: RE (%) | B: Test Error (%) | B: RE (%) |
|---|---|---|---|---|---|---|
| ViT-B/16(300M) | 86 | 15.85 | x | x | x | x |
| ViT-L/16(300M) | 307 | 12.84 | x | x | x | x |
| ViT-H/14(300M) | 632 | 11.45 | 11.39 | 0.5 | 11.39 | 0.5 |
| ViT-g/14(3B) | 1,010 | 10.34 | 10.54 | 1.9 | 10.54 | 1.9 |
| ViT-G/14(3B) | 1,840 | 9.55 | 9.36 | 2.0 | 9.54 | 0.1 |
| ViT-22B(3B) | 22,000 | 8.9 | 6.33 | 28.8 | 6.34 | 28.7 |

Table 11: Forward prediction of the top-1 test error for ViT models on ImageNet, pretrained on JFT. Parameter counts and error rates are taken from Dehghani et al. (2023). The absolute value of the slope is 1/6.042.

**Forward and Backward:** The parameter size of the last model (ViT-22B) is so much larger than that of the other models that using its data for prediction would require a long extrapolation, risking a loss of accuracy. Thus, the slope is calculated from the two largest remaining models: ViT-g/14 and ViT-G/14. This allows us to calculate the prediction error bidirectionally, and we use forward prediction to estimate the error for ViT-22B. We apply backward prediction to the rest of the other models.

| Model(JFT-) | Size (M) | Error (%) | A: Pred. Err. (%) | A: RE (%) | B: Test Error (%) | B: RE (%) |
|---|---|---|---|---|---|---|
| ViT-B/16(300M) | 86 | 15.85 | 15.2 | 4.1 | 14.32 | 9.6 |
| ViT-L/16(300M) | 307 | 12.84 | 12.6 | 1.8 | 12.11 | 5.6 |
| ViT-H/14(300M) | 632 | 11.45 | 11 | 3.9 | 11 | 3.9 |
| ViT-g/14(3B) | 1,010 | 10.34 | x | x | x | x |
| ViT-G/14(3B) | 1,840 | 9.55 | x | x | x | x |
| ViT-22B(3B) | 22,000 | 8.9 | 6.87 | 22.8 | 6.87 | 22.8 |

Table 12: Forward and backward prediction of the top-1 test error for ViT models on ImageNet, pretrained on JFT. Parameter counts and error rates are taken from Dehghani et al. (2023). The absolute value of the slope is 1/7.542.

