# OpenReview forum: "Model-Size Scaling Laws for Classification Networks via Manifold Geometry and Box-Model Dimension"
_TMLR — Under review for TMLR_

### Review · Reviewer_qMze · 2026-06-26

**Summary Of Contributions:**

This paper aims to provide a geometric explanation for scaling laws in classification networks. The central object of the analysis is the partition of the input space into the decision regions of the classifier. The generalization error is related to the size of such regions and their distance from the true decision boundary. This geometric picture has 3 main consequences.

First, the error scales as a power of the sample size, where the exponent is the inverse of the effective input dimension. Intuitively, this follows from requiring the training samples to densely cover the input domain.

Second, the error scales as a power of the sample size, where the exponent is the inverse of a “box-model dimension.” This quantity is analogous to the box-counting dimension for fractals, characterising how the number of model parameters must grow in order to refine the classifier’s partition of the input space. The resulting exponent is expressed through a model-dependent parameter $\alpha$. This part of the submission is difficult to assess in its current form: it is not clear what exactly #$W(s)$ denotes in the definition of the box-model dimension, whether it refers to a minimum over architectures, the parameter count of a constructed model, or the parameter count of a trained model. It is also unclear whether the parameter $\alpha$ is an independently measurable property of the architecture/learning method, or whether it must itself be inferred from the observed scaling-law exponent.

Third, the paper argues that modular neural architectures exhibit discrete scale invariance, resulting in log-periodic structure around the main power-law trend. The empirical section then uses ResNet, VGG, and Inception-family models to test whether the performance of larger models can be extrapolated from smaller models using the proposed scaling relation.

Overall, the paper offers an interesting attempt to connect scaling laws and the geometry of decision boundaries. However, in its current form, I found the main theoretical and empirical claims insufficiently clarified. The sample-size result is closely related to standard covering-number or curse-of-dimensionality arguments; the box-model dimension is not yet defined operationally enough to make the parameter-scaling claim convincing, especially because the role and measurability of $\alpha$ are unclear; and the empirical validation of discrete scale invariance appears to demonstrate power-law extrapolation (which is a direct consequence of scaling laws) rather than the distinctive consequences of the proposed discrete-scale-invariance mechanism.

**Audience:**

No

**Audience Explanation:**

The general direction of connecting neural scaling laws to the geometry of decision boundaries is relevant to TMLR. However, I am not convinced that the current findings, as stated, would yet be sufficiently informative for the TMLR audience. The sample-size result appears to recover a standard curse-of-dimensionality dependence on the effective input dimension (see e.g. refs. of [this paper](https://jmlr.org/papers/v18/14-546.html)). The parameter-size result suffers from the issues described in the previous section. Finally, the empirical results only demonstrate extrapolation from smaller to larger models---which was already demonstrated in the original Neural Scaling Laws papers---but do not clearly establish the specific discrete-scale-invariance mechanism claimed by the paper.

**Broader Impact Concerns:**

No further concerns

**Claims And Evidence:**

No

**Claims Explanation:**

Partially supported, as there is no evidence for the model-size scaling claim. The issue is not simply that the derivation is hard to follow, but that a key quantity needed for the claim is not specified in a way that would let a reader verify or apply the result. If $\alpha$ is a primitive quantity that can be independently determined, then the theory could predict the model-size exponent. If instead $\alpha$ is obtained by fitting the scaling law, then the argument does not explain the scaling law; it only rewrites it in different variables.

**Requested Changes:**

See above

---

> ### Author Response · Authors · 2026-07-16
> **Model-Size Scaling Laws for Classification Networks via Manifold Geometry and Box-Model Dimension**
>
> Dear Reviewer:
>
> Thank you for your comments. They have helped us a great deal in improving the manuscript, particularly your questions regarding the parameter $\alpha$. Your comments helped us improve Theorem 1 in the revised manuscript. Most of our updated discussion of $\alpha$ can be found in Section 4.1.2, particularly in the paragraphs following the theorem. Some implementation issues are discussed in Section 6, as highlighted there.
>
> We hope you understand that, because of the significant modifications to manifold geometry and the removal of the sample-size scaling law, we have changed the title to "Model-Size Scaling Laws for Classification Networks via Manifold Geometry and Box-Model Dimension" to make the manuscript more focused. Note that we use $W$ and \#W interchangeably throughout to denote the number of parameters.
>
> Best regards,
>
> Authors
>
> $\bullet$ **Summary Of Contributions (second and third):**
> We answer each directly.
>
> 1. What does \#W denote?
>
> \#W denotes the number of parameters of a specific trained model (i.e., a specific architecture together with a specific learning algorithm/training run), not a minimum over architectures.
>
> 2. Is $\alpha$ independently measurable, or must it be inferred from the observed scaling exponent?
>
> $\alpha^{\*}$ is defined implicitly in Eq.(20) (Theorem 1), so it depends jointly on $W$ and $\gamma_s$. Since $\gamma_s$ depends on both architecture and learning algorithm, so does $\alpha^{\*}$. In practice $\gamma_s$ cannot be measured directly (Section 6), so $\alpha^{\*}$ cannot be computed from a single point; instead we infer it by fitting two observed points (Eq.(35), Section 6.1). Thus $\alpha^*$ has a precise theoretical definition but must be inferred empirically, not measured independently.
>
> On the asymptotic regime: Theorem 1(iii) shows that, asymptotically, $\text{error} \propto W^{-1/\bar D_{\mathcal M}}$, where $\bar D_{\mathcal M} = \log_2(1+\alpha_{\mathcal M})$ is the box-model dimension, explaining the log-log linear behavior for large networks. For smaller networks (Theorem 1(i)), $\alpha^*$ has no simple closed form and must be determined numerically or as a superposition of components.  A special case is presented in Section 5: for architectures constructed by concatenating repeated modules (e.g., ResNet, VGG, ViT, InceptionNet), we show that the scaling law manifests as a log-periodic pattern superimposed on a power-law trend in log-log space. The scaling curve for such architectures decomposes into a superposition of two simple modes, one linear and one periodic; the linear component corresponds to the box-model dimension, while the periodic component is an additional oscillation that vanishes as the network becomes sufficiently large.
>
>
> $\bullet$ **Overall $\cdots$:** We address each of the three points raised.
>
> We address each of the three points raised.
>
> On the sample-size result: we have removed this section, as detailed above regarding the sample-size scaling proof to Reviewer ZM2e.
>
> On the box-model dimension and $\alpha$: for large networks, Theorem 1(iii) gives $\text{error} \propto W^{-1/\bar D_{\mathcal M}}$, explaining the log-log linear behavior widely observed empirically (see Section 2). For smaller networks, Theorem 1(i) shows the relationship is generally not log-log linear. $\alpha^{\*}$ is defined via $\gamma_s \cdot W^{1/\alpha^{\*}} = 1$, but since $\gamma_s$ cannot be measured directly, $\alpha^*$ must in practice be inferred from at least two observed points rather than computed at a single one; we agree this limits how operationally well-defined $\alpha$ currently is.
>
> On discrete scale invariance: we agree our results mainly demonstrate power-law extrapolation, not the distinct log-periodic signature. We have added a backward-prediction experiment (Section 6.1) as a consistency check. Confirming the log-periodic structure would require residual analysis across many model sizes, which published benchmarks do not provide enough points for; we state this limitation in Section 6.1 and leave it as future work.
>
> $\bullet$ **Are the claims $\cdots$:**
>
> $\alpha$ (equivalently, the box-model dimension $\bar D_{\mathcal M}$) is not, in general, primitively determinable ahead of observing the scaling curve: Theorem 1(iii) shows only that a log-log linear relationship holds asymptotically with slope $1/\bar D_{\mathcal M}$, not this slope's value. So $\alpha$ must be fit from data; our theory explains why the linear relationship holds in the log-log plane, not what its slope is.
>
>
>
> We have not established a fully independent computation of $\bar D_{\mathcal M}$ from architecture alone, since it also depends on the learning algorithm. We conjecture this dependence weakens in the interpolation zone; this remains illustrative rather than proven, and we discuss it as a limitation in Section 6.

---

### Review · Reviewer_W66D · 2026-06-27

**Summary Of Contributions:**

The paper aims to provide a theoretical explanation of scaling laws for classification networks. It derives generalization-error bounds based on the geometry of decision regions, using two main assumptions:1) training data form a dense covering of the input domain; 2) inside every decision region of the trained network, one has access to enough samples. Under a semi-algebraic assumption on the target classification boundary, the paper claims that the generalization error scales with sample size according to the effective dimension of the data manifold. The empirical evaluation fits the proposed scaling law using the two smallest models of each architecture family and extrapolates the performance of larger models using published benchmark results.

**Audience:**

No

**Audience Explanation:**

While the topic of neural scaling laws is certainly of interest, I do not believe the current manuscript provides findings that would be of sufficient interest to the TMLR audience. The theoretical contribution is largely a geometric analysis of abstract partitions under strong assumptions, and the connection to neural networks is not convincingly established. As a result, I do not think the paper advances our understanding of neural network scaling laws.

**Claims And Evidence:**

No

**Claims Explanation:**

1) **Weak connection with neural networks.**
The paper derives scaling based on some abstract geometric properties of the trained network. Any other classifier inducing decision regions with similar properties will lead to the same result. The contribution is primarily a geometric analysis of abstract partitions rather than a neural scaling law paper.

2) **Strong assumptions**
The first assumption is unrealistic in a high-dimensional setting. The authors claimed that all results stated in terms of $d$ carry over $d_e$. However, to my understanding, this does not resolve the curse of dimensionality but rather replaces it with a substantially stronger assumption on the data geometry. The paper does not justify why the covering arguments established in the ambient space remain valid after this substitution.

The second assumption is even more problematic. It assumes that every decision region of the trained network contains enough training samples for the empirical histogram to accurately approximate the underlying probability mass. Since these decision regions are themselves learned from the training data, they form a data-dependent partition rather than a fixed one. Standard concentration results, such as Hoeffding's inequality, do not directly apply in this adaptive setting.

Consequently, the connection with neural networks remains rather weak: the main results rely on strong assumptions about the geometry of the learned partition, but the paper does not explain why trained neural networks should satisfy these assumptions.


3) **Lack of rigour**
The paper also suffers from a lack of mathematical rigor. The main assumptions and the definition of key parameters such as $\gamma_s$ are stated mostly in informal language rather than as precise mathematical conditions, which makes it difficult to determine the exact scope of the results.

The related-work section is also incomplete for a paper claiming to explain neural scaling laws: it misses important parts of the recent scaling-law literature (there is numerous theoretical work on linear regression, such as Lin et al 2024, the sum of single index model Ren et al 2025, etc) and does not clearly position the contribution relative to existing theoretical and empirical work. Overall, the presentation gives the impression that the main claims are much stronger than what is actually proved. I do not believe the paper provides a "theoretical foundation for scaling laws" in classification networks.

3) **Experiments do not validate the assumptions**
Since the main purpose of the work is to provide scaling laws based on two strong assumptions, I think the authors should at least check experimentally whether these assumptions hold.

**Requested Changes:**

1) Better motivate the connection with neural networks.
2) Provide precise definitions of the mathematical objects and assumptions.
3) Provide empirical evidence that the main assumptions are satisfied, or at least approximately satisfied, in practice.

---

> ### Author Response · Authors · 2026-07-16
> **Model-Size Scaling Laws for Classification Networks via Manifold Geometry and Box-Model Dimension**
>
> Dear Reviewer:
>
> Your comments helped us improve the revised manuscript throughout. We have introduced three assumptions to support our sampling arguments: a manifold assumption, a distribution assumption, and a histogram-or-interpolation assumption. Together, these three assumptions control the sampling argument and the generalization error. With Lemmas 1 and 2 and simplification of notation in Section 3.1, we translate our results from the previous assumptions to this new set of assumptions. We believe the current assumptions are more acceptable to the machine learning community.
>
> Because of the significant modifications to manifold geometry and the removal of the sample-size scaling law, we have changed the title to "Model-Size Scaling Laws for Classification Networks via Manifold Geometry and Box-Model Dimension" to make the manuscript more focused.
> Best regards,
>
> Authors
>
>
> $\bullet$ **Weak connection with neural networks:**
>
>
> The neural-network connection enters specifically through how such networks approximate functions with increasing granularity as the number of activation functions grows (universal approximation theorem for shallow and deep neural networks, Cybenko 1989, Yarotsky 2017). A piecewise-linear network partitions the input space into decision regions, and stacking layers recursively refines this partition, with region count growing exponentially in depth. This connects to the well-studied problem of counting ReLU linear regions (Montúfar 2014, Serra 2018); combined with a manifold assumption, it lets us control the number of regions via a covering argument, avoiding the curse of dimensionality. Please see the highlighted part in the Introduction.
>
> The connection with neural networks is completed via the relationship between $\gamma_s$ and the parameter count #W (Lemma 3 through Theorem 1), which translates a geometric bound into a scaling law in terms of parameter count, the central object in neural scaling laws. An abstract classifier without a notion of parameter count would not admit this translation. Both connections are depicted in the fourth and fifth paragraphs in the Introduction.
>
> $\bullet$ **Strong assumptions:**
>
>
>
> We agree that our previous treatment did not, by itself, resolve the curse of dimensionality, and we have revised the manuscript to address this more carefully. We now adopt a manifold assumption together with two additional assumptions, detailed in Section 3.1, that together with Lemmas 1 and  2 allow data to occupy a thin neighborhood of the manifold (rather than lying on it exactly) while still controlling the resulting generalization error. Please refer to Section 3.1, and the two lemmas therein, to justify our claim.
>
> $\bullet$ **The second assumption:**
>
> The histogram assumption is hard to justify with a finite number of samples, though it holds as the sample size goes to infinity: under Assumption (bounded support), the histogram-based estimate $n_p/N$ is an unbiased estimator of the true regional mass $\int_p \mathcal P(x)\,dx$, with variance vanishing as $N\to\infty$ (Scott 2015, Rudemo 1982). Our last assumption also includes an interpolation case, commonly used in the study of scaling laws, which can be verified with a finite number of samples.
>
> On the connection with neural networks: for deep networks with piecewise-linear activations (e.g., ReLU), it is well established (Montúfar et al., 2014; Serra et al., 2018) that decision regions are finite unions of polyhedra, a simple special case of semi-algebraic sets, supporting the semi-algebraicity assumption underlying our stratification argument. However, these results give only upper bounds on the number of linear regions as a function of depth and width; the exact number and shape of regions for a specific trained network cannot be determined from these bounds alone. Since $\gamma_s$ and $\beta_l$ are derived from this partition, their true values are similarly hard to obtain numerically, and we cannot currently establish finer regularity properties such as positive reach $\tau$ or explicit bounds on $\gamma_s,\beta_l$. As noted in Section 6, we do not currently know how to verify or estimate these quantities for trained networks, and establishing more precise conditions is an important direction for future work.
>
> $\bullet$ **Lack of rigour:**
>
> The enclosing and inscribed radii $\gamma_s$ and $\beta_l$ are now formally defined in Definition 1 (Section 3.1)
>
> $\bullet$ **Experiments:**
>
> We discuss this difficulty explicitly in Section 6: the key geometric quantities, $\gamma_s$ and $\beta_l$, are derived from the decision regions induced by a trained network, and evaluating them requires determining the number of decision regions and delineating each region's boundary, both of which are computationally hard for a deep network. We do not currently know how to estimate these quantities even heuristically. We leave the experimental validation of these assumptions as an important direction for future work.

---

### Review · Reviewer_g6NB · 2026-07-03

**Summary Of Contributions:**

This paper proposes a geometric explanation for scaling laws in classification networks by relating generalization error to the granularity of the decision regions induced by a neural classifier. Under the assumption that the target classification boundary is semi-algebraic, and under dense-covering and histogram-approximation assumptions, the paper derives a generalization bound controlled by the smallest enclosing radius $\gamma_s$ of the learned decision regions near the target boundary. This leads to a sample-size scaling law whose exponent depends on the effective/intrinsic dimension $d_e$ of the data distribution. The paper also introduces the notion of a “box-model dimension,” intended to capture how the number of parameters required by a model family grows as the covering radius decreases, and uses it to derive a parameter-count scaling law whose exponent depends on architecture and learning method. As a secondary contribution, the paper argues that modular architectures may exhibit discrete scale invariance, leading to approximately linear behavior in log-log plots of error versus parameter count, and illustrates this idea using published results for ResNet, VGG, and Inception-style architectures on ImageNet, CIFAR-100, and Udacity. The main strength of the paper is that it attempts to give a geometric mechanism behind scaling laws rather than only fitting empirical power laws, and the distinction between sample-size scaling and architecture-dependent parameter scaling is conceptually interesting. The main weakness is that the connection between the theoretical quantities and practical trained networks remains indirect: the assumptions are strong, the empirical section does not directly measure $\gamma_s$, $\beta_l$	​, or the box-model dimension, and the experiments rely on a small number of previously reported benchmark points.

**Additional Comments:**

I appreciated the revised framing of the paper and found the geometric perspective interesting. One connection that could make the paper more broadly accessible is a short discussion of how the proposed ideas relate to current LLM scaling laws and transformer architectures. For example, transformers are also built from repeated modular blocks, so the discussion of discrete scale invariance may have natural parallels with depth/width/block-wise scaling in modern language models. Similarly, the proposed distinction between sample-size scaling, governed by intrinsic data dimension, and parameter-count scaling, governed by architecture-dependent quantities, seems relevant to how LLM performance changes with data, model size, and architecture. I do not think the paper needs to fully develop a transformer theory, but a brief discussion of these parallels would help readers place the contribution in the broader context of contemporary scaling-law research.

**Audience:**

Yes

**Audience Explanation:**

Yes. The paper addresses scaling laws, which are a topic of broad interest to the TMLR community, especially for researchers working on generalization, approximation theory, and empirical model scaling. The attempt to explain scaling behavior through decision-boundary geometry, intrinsic dimension, and architecture-dependent parameter scaling is conceptually interesting. Even if the current evidence is not fully convincing, the proposed perspective and the notion of a box-model dimension could be useful for researchers thinking about why power-law-like behavior appears in neural networks.

**Broader Impact Concerns:**

I do not see any major direct ethical concerns with this work. The paper is primarily theoretical and empirical, and does not introduce a new deployed system, dataset, or application that would create immediate societal risks. A minor point the authors could mention is that scaling-law predictions may influence decisions about training larger models, which has compute and environmental implications. At the same time, better scaling predictions could also help reduce wasted compute by allowing researchers to estimate performance before running very large experiments. A brief broader-impact statement noting this tradeoff would be sufficient.

**Claims And Evidence:**

No

**Claims Explanation:**

The paper presents an interesting theoretical framework and the revised manuscript is more careful about its assumptions and limitations, but the evidence is still only partially convincing. Although the empirical section includes several CNN families, the experiments mainly show that simple extrapolation works on a small set of published benchmark results; they do not directly measure the geometric quantities central to the theory, such as $\gamma_s$, $\beta_l$, boundary-tube volume, or box-model dimension. Thus, the experiments support the plausibility of the proposed scaling behavior, but they do not fully validate the claimed geometric mechanism behind it.

**Requested Changes:**

Critical to acceptance: I think the paper would benefit from presenting the empirical results in a slightly more conservative way. The current experiments are interesting and cover several CNN families, but they mainly demonstrate that the proposed extrapolation method can work on published benchmark results. To make the connection to the theory clearer, I would encourage the authors to explicitly state that these experiments are illustrative evidence for the proposed mechanism, rather than a direct validation of all the underlying geometric quantities. It would also be helpful to add a short discussion clarifying how quantities such as $\gamma_s$, $\beta_l$, boundary-tube volume, and box-model dimension might be estimated or approximated in practical trained networks, or to acknowledge this as an important direction for future work.

Would strengthen the work: The paper would be stronger with additional controlled experiments using fixed training protocols, multiple random seeds, and more model sizes. A comparison against standard power-law or broken-power-law baselines would also help show the added value of the proposed prediction method. Since the paper discusses discrete scale invariance, it would be useful to analyze residuals around the fitted power law to see whether the predicted log-periodic structure appears in practice. These additions are not necessary to appreciate the main idea, but they would make the empirical support more convincing and help readers better understand the practical implications of the theory.

---

> ### Author Response · Authors · 2026-07-16
> **Model-Size Scaling Laws for Classification Networks via Manifold Geometry and Box-Model Dimension**
>
> Dear Reviewer:
>
> Thank you for your comments. Your constructive suggestions helped us address a key weakness: the indirect relation between theory and experiments. Based on your comments, we have reorganized the manuscript to focus on the theoretical contribution, and to investigate the difficulties that arise in verification experiments. In particular, Section 6 now contains the limitations and future directions. We hope that in the future we can carry out your experimental suggestions carefully.
>
> In response to comments from all reviewers, we have made significant modifications related to manifold geometry and removed the sample-size scaling law; accordingly, we have changed the title to "Model-Size Scaling Laws for Classification Networks via Manifold Geometry and Box-Model Dimension" to make the manuscript more focused.
>
> Best regards,
>
> Authors
>
> $\bullet$ **Summary Of Contributions:**
>
> We now emphasize that the theoretical contribution is our primary contribution. Architectures built by concatenating repeated modules (e.g., stacked residual or convolutional blocks) may yield a more easily computable box-model dimension in future work, since such repeated structure may lead to a self-similar decision-region geometry more analytically tractable than that of an arbitrary trained network. Predicting scaling behavior may benefit from such a design, since discrete scale invariance provides an analytical basis for justifying predictions, as an alternative to relying solely on experiments.
>
> We agree there is a gap between numerical verification and our theory. Section 6 explains $\gamma_s$ and $\beta_l$ cannot be easily obtained (highlighted). We also acknowledge the benchmark examples in Section 6.1 provide insufficient data to verify discrete scale invariance. We address both points as future work (Section 6).
>
> $\bullet$ **Are the claims:**
>
> Our experiments support the plausibility of the proposed scaling behavior but do not directly validate the underlying geometric mechanism. We agree that directly measuring $\gamma_s$, $\beta_l$, and the boundary-tube volume would be needed to numerically verify our theory; however, we do not currently know how to estimate these quantities for a trained one-hot network, since they depend on the geometry of the network's induced decision regions, which is difficult to characterize directly (Section 6). We also agree that we do not provide sufficient empirical evidence specifically for the box-model dimension. As one direction toward closing this gap, a network design explicitly satisfying the discrete scale-invariance property could produce a log-log linear relationship whose slope is directly related to the box-model dimension, offering a more targeted empirical test than extrapolation from existing benchmarks; we hope to pursue this in future work (Section 6).
>
> $\bullet$ **Critical to acceptance:**
>
> We agree, and now state explicitly in the revised manuscript that our experiments are illustrative evidence for the proposed geometric mechanism, not a direct validation of the underlying geometric quantities.
>
> Section 6 discusses the numerical difficulties in evaluating $\gamma_s$, $\beta_l$, and the boundary-tube volume for a trained network: counting the number of decision regions and delineating each region's boundary are both computationally hard, and we do not currently know how to estimate these quantities even heuristically.
>
> We also acknowledge insufficient numerical evidence for our conclusion that, for a sufficiently large network, the scaling law is characterized asymptotically by the box-model dimension (Theorem 1(iii)). As one path toward addressing this without relying on asymptotic analysis alone, we propose an architecture built by repeatedly concatenating identical modules, which yields an analytically tractable box-model dimension. Since many practical architectures are already modular, we use published benchmarks to illustrate this idea, while acknowledging these benchmarks were not reported for this purpose; substantially more experiments are needed to fully justify the approach, which we leave as future work (Section 6).
>
> In Section 6.1 and Appendix F, we add a backward-prediction result (estimating smaller models' performance from larger ones) to further examine consistency, and a new ViT benchmark to test whether the module-based perspective extends beyond convolutional architectures.
>
>
> **Would strengthen the work:**
> These suggestions would meaningfully strengthen the empirical support for our proposed mechanism and help justify discrete scale invariance empirically. We have not yet carried out controlled experiments with fixed training protocols and multiple seeds, comparisons against power-law or broken-power-law baselines, or residual analyses of the predicted log-periodic structure; we leave these for future work (Section 6).
>
> \bullet$ **Additional Comments:**
>
> We have included this suggestion; see Sections 1 and 5 (both highlighted).

---

### Review · Reviewer_ZM2e · 2026-07-07

**Summary Of Contributions:**

The submission aims to provide a theoretical explanation of empirical scaling laws for classification networks. Its central mechanism is geometric: a one-hot classifier partitions the input domain into decision regions; when these regions are sufficiently small, classification errors are argued to occur only near the target decision boundary. The paper then derives two scaling claims:

- **Sample-size scaling:** the generalization error bound scales approximately as $N^{-1/d_e}$, where $d_e$ is an effective or intrinsic dimension of the data distribution.
- **Model-size scaling:** the generalization error bound scales approximately as $ W^{-1/D_M}$, where $D_M$ is a proposed “box-model dimension” describing how parameter count grows as decision-region covering radius decreases.

The paper also introduces a discrete scale-invariance interpretation for modular architectures and gives preliminary empirical demonstrations using publicly reported ResNet, VGG, and InceptionNet results.

The main conceptual idea—relating generalization error to the probability mass of a tubular neighborhood around the target decision boundary—is interesting. However, the current manuscript has major technical gaps in its central lemmas, theorem proofs, assumptions, and empirical validation.

**Audience:**

Yes

**Audience Explanation:**

The broad idea of linking scaling laws to decision-region geometry and boundary tubes may interest researchers in generalization theory, neural scaling laws, geometric learning theory, and expressivity. The proposed box-model dimension could also be conceptually useful if reformulated rigorously.

However, TMLR requires interesting claims to be supported by accurate, convincing, and clear evidence. The current version does not meet that standard. The paper may provide useful intuition, but the formal results and empirical evidence are too weak for acceptance.

**Broader Impact Concerns:**

No concerns

**Claims And Evidence:**

No

**Claims Explanation:**

The paper makes broad claims about sample-size scaling, model-size scaling, and extrapolation from small to large models. These claims are not adequately supported. The problems are not minor presentation issues; they affect the correctness of the main results.



- Lemma 2 is the paper’s central result. It attempts to show that the expected classification error is bounded by the empirical training error plus a tubular-neighborhood term around the classification boundary. The high-level intuition is plausible, but the proof has multiple technical defects. First, in Appendix B, the proof introduces
$h_f(x;\|x-y\|_2)=g(f(x),f(y)),$ but $h_f$ is not properly defined as there could be two same points lying in the ball centered at x with radius $\eta_f(x)$ but having different labels.
Due to the same reason, $\eta_f(x)$'s definition has problems, it should be defined as the distance between x and the boundary of f.


- Eq. (31) effectively assumes $\mathcal L^d(T_r(S))
\le
\omega_{d-k}r^{d-k}\mathcal H^k(S)$ for each $k$-dimensional stratum $S$. This inequality is not true in general. A simple counterexample is a line segment $S\subset\mathbb R^2$ of length $L$. Its $r$-neighborhood has area $2rL+\pi r^2,$ whereas the claimed leading term is only $2rL$. The inequality fails by the endpoint contribution $\pi r^2$. For curved manifolds, curvature terms also appear. This also leads to the wrong Figure.3, as there should be two half-balls with the cylinder.

- The paper states $c_P\in(0,1)$ and interprets it as an average density over the boundary tube. This is generally incorrect. If $p$ is a density with respect to Lebesgue measure, the average density over a measurable set $A_r$ is $c_P(r)=\frac{\int_{A_r}p(x)\,dx}{\mathcal L^d(A_r)}.$ This quantity can be zero, can exceed one, and generally depends on $r=2\gamma_s$.

- The proof of Theorem 1 / Appendix C does not establish the claimed $N^{-1/d_e}$ scaling. The paper’s sample-size scaling proof is substantially flawed. The proof defines the expected number of samples inside a ball as $t_N(x)=N P(B(x,\beta_l)).$ Using a lower density assumption, it obtains
$t_N(x)\ge cN\beta_l^{d_e}.$ The proof then takes $t_N(x)=1$ as the “minimal case” and concludes $\beta_l\lesssim N^{-1/d_e}.$
This is not a valid implication. For fixed $x$, fixed network $M$, and fixed $\beta_l$, the probability $P(B(x,\beta_l))$ is fixed, so $t_N(x)=NP(B(x,\beta_l))$ grows linearly with $N$. One cannot set $t_N(x)=1$ unless one is redefining $\beta_l$ as a new radius depending on $N$. This step is a heuristic critical-scale calculation, not a proof.
Moreover, the proof uses $\gamma_s\le K\beta_l$, but this is a strong shape-regularity assumption on network decision regions if the network decision region changes with samples. It does not follow from boundedness of $\gamma_s$ and $\beta_l$, especially in an asymptotic regime where both may go to zero. Decision regions can become needle-like, with $\gamma_s/\beta_l\to\infty$.

- Like above, from the theoretical perspective, the presentation and proofs are generally poor without explicit statement of the problem and the full assumptions/definitions, while some given assumptions are too strong.

- The empirical section uses publicly reported model sizes and test errors for ResNet, VGG, and InceptionNet, then fits or extrapolates power laws using very few points. This is not adequate evidence for the theoretical mechanism. The experiments do not measure or estimate: the decision-region radius $\gamma_s$, the inscribed radius $\beta_l$ and whether the proposed box-model dimension corresponds to actual trained network partitions. Moreover, test error is used as a proxy for the generalization bound, but the theory concerns expected test error minus training error or an upper bound thereof. Several experiments use only two or three model sizes to estimate scaling exponents, which is not enough to substantiate the discrete scale-invariance claim.

**Requested Changes:**

See above. There are also some minor issues:

1. The paper repeatedly writes $f=\bigcup_i g(i)$, but $f$ is a function and $g(i)$ is a set. The correct statement is that the domain is partitioned into class regions $G_i=f^{-1}(e_i)$.

2. The phrase “semi-algebraic sets can realize arbitrarily complicated shapes” is misleading. Semi-algebraic sets are tame and exclude infinite oscillation and fractal phenomena. They can have arbitrarily high finite algebraic complexity, but not arbitrary pathological boundaries.

3. Rigorously state the problem and related definitions.

---

> ### Author Response · Authors · 2026-07-16
> **Model-Size Scaling Laws for Classification Networks via Manifold Geometry and Box-Model Dimension**
>
> Dear Reviewer:
>
> We sincerely thank you for identifying errors. This feedback substantially strengthened the manuscript.
>
> We regard the main contribution of this manuscript as theoretical, so the correctness of the analysis is paramount, and we have addressed your comments on it accordingly.
>
> We have made significant modifications. As a result, we have changed the title to make the manuscript more focused.
>
> Best regards,
>
> Authors
>
> $\bullet$ **$h_f(x; |x-y|_2) = g(f(x), f(y))$:**
> Lemma 2 in the previous submission is Lemma 4 in the revised submission
>
> We define $\eta_f(x)$ explicitly as the distance from $x$ to the
> decision boundary of its class region (Eq(45)). We have highlighted the relevant paragraph in Appendix C.
>
> $\bullet$ **$\mathcal L^d(T_r(S)) \leq \omega_{d-l} r^{d-k} \mathcal H^k(S)$:**
> Lemma 5 now includes an explicit correction term accounting for the frontier $\partial f_k^i$ of each stratum (Eq.(58)); the second term on the right-hand side reflects that each frontier point contributes a half-ball cap rather than a full ball.
> We have propagated this correction term through Eq.(60) to the subsequent bound preceding Eq.(61). We note that this correction term is one order higher in $\gamma_s$ than the leading term of the same stratum, and does not affect the leading-order approximation used in the subsequent scaling-law derivation, since the top-dimensional stratum $|f_{d-1}|$ continues to dominate as $\gamma_s \to 0$ (Lemma 4(ii)).
>
> The curvature contribution is controlled by the reach condition $r \leq \tau(f_k^i)$ already present in Lemma 5 and Definition 3, which prevents the tube volume from exceeding the flat estimate due to curvature alone.
>
> We have removed the previous Figure 3. Please refer to the proof of Lemma 4 for the detailed derivation.
>
> $\bullet$ **The paper states $c_{\mathcal P} \in (0,1)$**
>
> As in our "Simplification of notation" paragraph, we set $\delta = 0$ throughout, so that Assumption (bounded support) reduces to: $\mathcal P$ is supported on $\mathcal X$, with $k \leq \mathcal P(x) \leq K$ for all $x \in \mathcal X$, for constants $0 < k \leq K < \infty$. Under the same simplification, $\mathcal X$ is identified with our input domain $B_d(\mathbf{0},1)$ via the parametrization of (Niyogi2008), so the density bounds $k \leq \mathcal P(x) \leq K$ hold for all $x \in B_d(\mathbf{0},1)$. Since $A(2\gamma_s, \partial f) \subseteq B_d(\mathbf{0},1)$, the pointwise density bounds hold in particular on $A(2\gamma_s,\partial f)$, and integrating them directly gives Eq. (62), so $c_{\mathcal P} \in [k, K]$. This holds for every $r = 2\gamma_s$ satisfying our stated assumptions, since it follows directly from the pointwise bounds on $\mathcal P$ rather than any assumption specific to a particular value of $r$.
>
> $\bullet$ **assumptions are too strong:**
>
> We have revised our assumption. Please refer to Section 3.1 for assumptions, lemmas, and simplification of notation.
>
>
> $\bullet$ **Empirical section:**
>
> On using test error as a proxy for the generalization gap: our theory covers histogram-approximation and interpolation regimes. The former is hard to verify empirically; our benchmarks (ResNet, VGG, InceptionNet, ViT) instead rely on the interpolation regime, since such over-parameterized architectures are widely observed to reach near-zero training error, though the original sources do not report it directly. This is an assumption, not a confirmed property of the cited benchmarks, and we explicitly note this limitation.
>
> The current experiments serve an illustrative rather than confirmatory role. Our primary contribution is theoretical; the empirical section (Section 6.1) is secondary and illustrative, using published benchmarks to suggest that discrete scale invariance could produce an approximately linear $\ln$--$\ln$ relationship between generalization error and parameter count. We agree that this alone does not substantiate the discrete-scale-invariance claim. Lacking an efficient estimator for their manifold-geometric definitions, we do not measure $\gamma_s$, $\beta_l$, or the box-counting dimension for trained networks.
>
> The current experiments serve an illustrative rather than confirmatory role. Our primary contribution is theoretical; the empirical section (Section 6.1) is secondary and illustrative, using published benchmarks to suggest that discrete scale invariance could produce an approximately linear $\ln$--$\ln$ relationship between generalization error and parameter count. We agree that this alone does not substantiate the discrete-scale-invariance claim. Because of a lack of an efficient estimation method given their manifold-geometric definition, we do not measure $\gamma_s$, $\beta_l$, or the box-counting dimension for actual trained networks. These are flagge as future work (Section 6).
>
> **Requested Changes:** The two minor issues have been corrected in the revised manuscript (both highlighted in Lemma 4 and a paragraph above the lemma).